# A Stochastic Composite Gradient Method with Incremental Variance Reduction

**Junyu Zhang**
University of Minnesota
Minneapolis, Minnesota 55455
zhan4393@umn.edu

**Lin Xiao**
Microsoft Research
Redmond, Washington 98052
lin.xiao@microsoft.com

## Abstract

We consider the problem of minimizing the composition of a smooth (nonconvex) function and a smooth vector mapping, where the inner mapping is in the form of an expectation over some random variable or a finite sum. We propose a stochastic composite gradient method that employs an incremental variance-reduced estimator for both the inner vector mapping and its Jacobian. We show that this method achieves the same orders of complexity as the best known first-order methods for minimizing expected-value and finite-sum nonconvex functions, despite the additional outer composition which renders the composite gradient estimator biased. This finding enables a much broader range of applications in machine learning to benefit from the low complexity of incremental variance-reduction methods.

## 1 Introduction

In this paper, we consider stochastic composite optimization problems

$$\underset{x \in \mathbf{R}^d}{\text{minimize}} \quad f\big(\mathbf{E}_\xi[g_\xi(x)]\big) + r(x), \tag{1}$$

where $f : \mathbf{R}^p \to \mathbf{R}$ is a smooth and possibly nonconvex function, $\xi$ is a random variable, $g_\xi : \mathbf{R}^d \to \mathbf{R}^p$ is a smooth vector mapping for $a.e.\ \xi$, and $r$ is convex and lower-semicontinuous. A special case we will consider separately is when $\xi$ is a discrete random variable with uniform distribution over $\{1, 2, \ldots, n\}$. In this case the problem is equivalent to a deterministic optimization problem

$$\underset{x \in \mathbf{R}^d}{\text{minimize}} \quad f\left(\frac{1}{n} \sum_{i=1}^{n} g_i(x)\right) + r(x). \tag{2}$$

The formulations in (1) and (2) cover a broader range of applications than classical stochastic optimization and empirical risk minimization (ERM) problems where each $g_\xi$ is a scalar function ($p = 1$) and $f$ is the scalar identity map. Interesting examples include the policy evaluation in reinforcement learning (RL) [e.g., 30], the risk-averse mean-variance optimization ([e.g., 28, 29], through a reformulation by [35]), the stochastic variational inequality ([e.g., 12, 15] through a reformulation in [10]), the 2-level composite risk minimization problems [7], etc.

For the ease of notation, we define

$$g(x) := \mathbf{E}_\xi[g_\xi(x)], \qquad F(x) := f(g(x)), \qquad \Phi(x) := F(x) + r(x). \tag{3}$$

In addition, let $f'$ and $F'$ denote the gradients of $f$ and $F$ respectively, and $g'_\xi(x) \in \mathbf{R}^{p \times d}$ denote the Jacobian matrix of $g_\xi$ at $x$. Then we have

$$F'(x) = \nabla\Big(f\big(\mathbf{E}_\xi[g_\xi(x)]\big)\Big) = \Big(\mathbf{E}_\xi[g'_\xi(x)]\Big)^T f'\big(\mathbf{E}_\xi[g_\xi(x)]\big).$$

Table 1: Sample complexities of CIVR (Composite Incremental Variance Reduction)

| Problem | Assumptions (common: $f$ and $g_\xi$ Lipschitz and smooth, thus $F$ smooth) | | |
| | $F$ nonconvex $r$ convex | $F$ $\nu$-gradient dominant $r \equiv 0$ | $F$ convex, $r$ convex $\Phi$ $\mu$-optimally strongly convex |
|---|---|---|---|
| (1) | $O\big(\epsilon^{-3/2}\big)$ | $O\big((\nu\epsilon^{-1})\log\epsilon^{-1}\big)$ | $O\big((\mu^{-1}\epsilon^{-1})\log\epsilon^{-1}\big)$ |
| (2) | $O\big(\min\{\epsilon^{-3/2}, n^{1/2}\epsilon^{-1}\}\big)$ | $O\big((n+\nu n^{1/2})\log\epsilon^{-1}\big)$ | $O\big((n+\mu^{-1}n^{1/2})\log\epsilon^{-1}\big)$ |

In practice, computing $F'(x)$ exactly can be very costly if not impossible. Due to the nonlinearity of the outer composition, simply multiplying the unbiased estimators $\mathbf{E}[\tilde{g}(x)] = g(x)$ and $\mathbf{E}[\tilde{g}'(x)] = g'(x)$ results in a biased estimator for $F'(x)$, namely, $\mathbf{E}[[\tilde{g}'(x)]^T f'(\tilde{g}(x))] \neq F'(x)$, see [e.g., 35]. This is in great contrast to the classical stochastic optimization problem

$$\operatorname*{minimize}_{x\in\mathbf{R}^d}\quad \mathbf{E}_\xi\big[g_\xi(x)\big] + r(x), \qquad (4)$$

where one can always get an unbiased gradient estimator for the smooth part. This fact makes such composition structure be of independent interest for research on stochastic and randomized algorithms.

In this paper, we develop an efficient stochastic composite gradient method called CIVR (Composite Incremental Variance Reduction), for solving problems of the forms (1) and (2). We measure efficiency by the sample complexity of the individual functions $g_\xi$ and their Jacobian $g'_\xi$, i.e., the total number of times they need to be evaluated at some point, in order to find an $\epsilon$-approximate solution. For nonconvex functions, an $\epsilon$-approximate solution is some random output of the algorithm $\bar{x} \in \mathbf{R}^d$ that satisfies $\mathbf{E}[\|\mathcal{G}(\bar{x})\|^2] \leq \epsilon$, where $\mathcal{G}(\bar{x})$ is the *proximal gradient mapping* of the objective function $\Phi$ at $\bar{x}$ (see details in Section 2). If $r \equiv 0$, then $\mathcal{G}(\bar{x}) = F'(\bar{x})$ and the criteria for $\epsilon$-approximation becomes $\mathbf{E}[\|F'(\bar{x})\|^2] \leq \epsilon$. If the objective $\Phi$ is convex, we require $\mathbf{E}[\Phi(\bar{x}) - \Phi^\star] \leq \epsilon$ where $\Phi^\star = \inf_x \Phi(x)$. For smooth *and* convex functions, these two notions are compatible, meaning that the dependence of the sample complexity on $\epsilon$ in terms of both notions are of the same order.

Table 1 summarizes the sample complexities of the CIVR method under different assumptions obtained in this paper. We can define a condition number $\kappa = O(\nu)$ for $\nu$-gradient dominant functions and $\kappa = O(1/\mu)$ for $\mu$-optimally strongly convex functions, then the complexities become $O\big((\kappa\epsilon^{-1})\log\epsilon^{-1}\big)$ and $O\big((n+\kappa n^{1/2})\log\epsilon^{-1}\big)$ for (1) and (2) respectively. In order to better position our contributions, we next discuss related work and then putting these results into context.

## 1.1 Related Work

We first discuss the nonconvex stochastic optimization problem (4), which is a special cases of (1). When $r \equiv 0$ and $g(x) = \mathbf{E}_\xi[g_\xi(x)]$ is smooth, Ghadimi and Lan [9] developed a randomized stochastic gradient method with iteration complexity $O(\epsilon^{-2})$. Allen-Zhu [2] obtained $O(\epsilon^{-1.625})$ with additional second-order guarantee. There are also many recent works on solving its finite-sum version

$$\operatorname*{minimize}_{x\in\mathbf{R}^d}\quad \frac{1}{n}\sum_{i=1}^n g_i(x) + r(x), \qquad (5)$$

which is also a special case of (2). By extending the variance reduction techniques SVRG [13, 34] and SAGA [6] to nonconvex optimization, Allen-Zhu and Hazan [3] and Reddi et al. [24, 25, 26] developed randomized algorithms with sample complexity $O(n + n^{2/3}\epsilon^{-1})$. Under additional assumptions of gradient dominance or strong convexity, they obtained sample complexity $O((n+\kappa n^{2/3})\log\epsilon^{-1})$, where $\kappa$ is a suitable condition number. Allen-Zhu [1] and Lei et al. [17] obtained $O\big(\min\{\epsilon^{-5/3}, n^{2/3}\epsilon^{-1}\}\big)$.

Based on a new variance reduction technique called SARAH [21], Nguyen et al. [22] and Pham et al. [23] developed nonconvex extensions to obtain sample complexities $O(\epsilon^{-3/2})$ and $O(n + n^{1/2}\epsilon^{-1})$ for solving the expectation and finite-sum cases respectively. Fang et al. [8] introduced another variance reduction technique called SPIDER, which can be viewed as a more general variant of SARAH. They obtained sample complexities $O(\epsilon^{-3/2})$ and $O\big(\min\{\epsilon^{-3/2}, n^{1/2}\epsilon^{-1}\}\big)$ for the two cases respectively, but require small step sizes that are proportional to $\epsilon$. Wang et al. [33] extended SPIDER to obtain the same complexities with constant step sizes and $O\big((n + \kappa^2)\log\epsilon^{-1}\big)$ under the gradient-dominant condition. In addition, Zhou et al. [36] obtained similar results using a nested SVRG approach.

In addition to the above works on solving special cases of (1) and (2), there are also considerable recent works on a more general, two-layer stochastic composite optimization problem

$$\underset{x \in \mathbf{R}^d}{\text{minimize}} \quad \mathbf{E}_\nu \big[ f_\nu \left( \mathbf{E}_\xi [g_\xi(x)] \right) \big] + r(x), \tag{6}$$

where $f_\nu$ is parametrized by another random variables $\nu$, which is independent of $\xi$. For the case $r \equiv 0$, Wang et al. [31] derived algorithms to find an $\epsilon$-approximate solution with sample complexities $O(\epsilon^{-4})$, $O(\epsilon^{-3.5})$ and $O(\epsilon^{-1.25})$ for the smooth nonconvex case, smooth convex case and smooth strongly convex case respectively. For nontrivial convex $r$, Wang et al. [32] obtained improved sample complexity of $O(\epsilon^{-2.25})$, $O(\epsilon^{-2})$ and $O(\epsilon^{-1})$ for the three cases mentioned above respectively.

As a special case of (6), the following finite-sum problem also received significant attention:

$$\underset{x \in \mathbf{R}^d}{\text{minimize}} \quad \frac{1}{m} \sum_{j=1}^{m} f_j \left( \frac{1}{n} \sum_{i=1}^{n} g_i(x) \right) + r(x). \tag{7}$$

When $r \equiv 0$ and the overall objective function is strongly convex, Lian et al. [19] derived two algorithms based on the SVRG scheme to attain sample complexities $O((m + n + \kappa^3) \log \epsilon^{-1}))$ and $O((m + n + \kappa^4) \log \epsilon^{-1}))$ respectively, where $\kappa$ is some suitably defined condition number. Huo et al. [11] also used the SVRG scheme to obtain an $O(m + n + (m + n)^{2/3} \epsilon^{-1})$ complexity for the smooth nonconvex case and $O((m + n + \kappa^3) \log \epsilon^{-1}))$ for strongly convex problems with nonsmooth $r$. More recently, Zhang and Xiao [35] proposed a composite randomized incremental gradient method based on the SAGA estimator [6], which matches the best known $O(m + n + (m + n)^{2/3} \epsilon^{-1})$ complexity when $F$ is smooth and nonconvex, and obtained an improved complexity $O((m + n + \kappa(m + n)^{2/3}) \log \epsilon^{-1})$ under either gradient dominant or strongly convex assumptions. When applied to the special cases (1) and (2) we focus on in this paper ($m = 1$), these results are strictly worse than ours in Table 1.

## 1.2 Contributions and Outline

We develop the CIVR method by extending the variance reduction technique of SARAH [21–23] and SPIDER [8, 33] to solve the composite optimization problems (1) and (2). The complexities of CIVR in Table 1 match the best results for solving the non-composite problems (4) and (5), despite the additional outer composition and the composite-gradient estimator always being biased. In addition:

- By setting $f$ and $g_\xi$'s to be the identity mapping and scalar mappings respectively, problem (2) includes problem (5) as a special case. Therefore, the lower bounds in [8] for the non-composite finite-sum optimization problem (5) indicates that our $O\big(\min\{\epsilon^{-3/2}, n^{1/2}\epsilon^{-1}\}\big)$ complexity for solving the more general composite finite-sum problem (2) is near-optimal.

- Under the assumptions of gradient dominance or strong convexity, the $O\left((n + \kappa n^{1/2}) \log \epsilon^{-1}\right)$ complexity only appeared for the special case (5) in the recent work [18].

Our results indicate that the additional smooth composition in (1) and (2) does not incur higher complexity compared with (4) and (5), despite the difficulty of dealing with biased estimators. We believe these results can also be extended to the two-layer problems (6) and (7), by replacing $n$ with $m + n$ in Table 1. But the extensions require quite different techniques and we will address them in a separate paper.

The rest of this paper is organized as follows. In Section 2, we introduce the CIVR method. In Section 3, we present convergence results of CIVR for solving the composite optimization problems (1) and (2) and the required parameter settings. Better complexities of CIVR under the gradient-dominant and optimally strongly convex conditions are given in Section 4. In Section 5, we present numerical experiments for solving a risk-averse portfolio optimization problem (5) on real-world datasets.

## 2 The composite incremental variance reduction (CIVR) method

With the notations in (3), we can write the composite stochastic optimization problem (1) as

$$\underset{x \in \mathbf{R}^d}{\text{minimize}} \quad \{\Phi(x) = F(x) + r(x)\}, \tag{11}$$

where $F$ is smooth and $r$ is convex. The proximal operator of $r$ with parameter $\eta$ is defined as

$$\mathbf{prox}_r^\eta(x) := \underset{y}{\text{argmin}} \left\{ r(y) + \frac{1}{2\eta} \|y - x\|^2 \right\}. \tag{12}$$

---
**Algorithm 1:** Composite Incremental Variance Reduction (CIVR)
---

**input:** initial point $x_0^1$, step size $\eta > 0$, number of epochs $T \geq 1$, and a set of triples $\{\tau_t, B_t, S_t\}$
    for $t = 1, \ldots, T$, where $\tau_t$ is the epoch length and $B_t$ and $S_t$ are sample sizes in epoch $t$.

**for** $t = 1, \ldots, T$ **do**
    Sample a set $\mathcal{B}_t$ with size $B_t$ from the distribution of $\xi$, and construct the estimates

$$y_0^t = \frac{1}{B_t} \sum_{\xi \in \mathcal{B}_t} g_\xi(x_0^t), \qquad z_0^t = \frac{1}{B_t} \sum_{\xi \in \mathcal{B}_t} g_\xi'(x_0^t), \tag{8}$$

    Compute $\tilde{\nabla} F(x_0^t) = (z_0^t)^T f'(y_0^t)$ and update: $x_1^t = \mathbf{prox}_r^\eta \left( x_0^t - \eta \tilde{\nabla} F(x_0^t) \right)$.

    **for** $i = 1, \ldots, \tau_t - 1$ **do**
        Sample a set $\mathcal{S}_i^t$ with size $S_t$ from the distribution of $\xi$, and construct the estimates

$$y_i^t = y_{i-1}^t + \frac{1}{S_t} \sum_{\xi \in \mathcal{S}_i^t} \left( g_\xi(x_i^t) - g_\xi(x_{i-1}^t) \right), \tag{9}$$

$$z_i^t = z_{i-1}^t + \frac{1}{S_t} \sum_{\xi \in \mathcal{S}_i^t} \left( g_\xi'(x_i^t) - g_\xi'(x_{i-1}^t) \right). \tag{10}$$

        Compute $\tilde{\nabla} F(x_i^t) = (z_i^t)^T f'(y_i^t)$ and update: $x_{i+1}^t = \mathbf{prox}_r^\eta \left( x_i^t - \eta \tilde{\nabla} F(x_i^t) \right)$.
    **end**
    Set $x_0^{t+1} = x_{\tau_t}^t$.
**end**
**output:** $\bar{x}$ randomly chosen from $\left\{ x_i^t \right\}_{i=0,\ldots,\tau_t-1}^{t=1,\ldots,T}$.

---

We assume that $r$ is relatively simple, meaning that its proximal operator has a closed-form solution or can be computed efficiently. The proximal gradient method [e.g., 20, 4] for solving problem (11) is

$$x^{t+1} = \mathbf{prox}_r^\eta \left( x^t - \eta F'(x^t) \right), \tag{13}$$

where $\eta$ is the step size. The *proximal gradient mapping* of $\Phi$ is defined as

$$\mathcal{G}_\eta(x) \triangleq \frac{1}{\eta} \left( x - \mathbf{prox}_r^\eta \left( x - \eta F'(x) \right) \right). \tag{14}$$

As a result, the proximal gradient method (13) can be written as $x^{t+1} = x^t - \eta \, \mathcal{G}(x^t)$. Notice that when $r \equiv 0$, $\mathbf{prox}_r^\eta(\cdot)$ becomes the identity mapping and we have $\mathcal{G}_\eta(x) \equiv F'(x)$ for any $\eta > 0$.

Suppose $\bar{x}$ is generated by a randomized algorithm. We call $\bar{x}$ an $\epsilon$-stationary point in expectation if

$$\mathbf{E}\left[ \|\mathcal{G}_\eta(\bar{x})\|^2 \right] \leq \epsilon. \tag{15}$$

(We assume that $\eta$ is a constant that does not depend on $\epsilon$.) As we mentioned in the introduction, we measure the efficiency of an algorithm by its sample complexity of $g_\xi$ and their Jacobian $g_\xi'$, i.e., the total number of times they need to be evaluated, in order to find a point $\bar{x}$ that satisfies (15). Our goal is to develop a randomized algorithm that has low sample complexity.

We present in Algorithm 1 the Composite Incremental Variance Reduction (CIVR) method. This methods employs a two time-scale variance-reduced estimator for both the inner function value of $g(\cdot) = \mathbf{E}_\xi[g_\xi(\cdot)]$ and its Jacobian $g'(\cdot)$. At the beginning of each outer iteration $t$ (each called an epoch), we construct a relatively accurate estimate $y_0^t$ for $g(x_0^t)$ and $z_0^t$ for $g'(x_0^t)$ respectively, using a relatively large sample size $B_t$. During each inner iteration $i$ of the $t$th epoch, we construct an estimate $y_i^t$ for $g(x_i^t)$ and $z_i^t$ for $g'(x_i^t)$ respectively, using a smaller sample size $S_t$ *and* incremental corrections from the previous iterations. Note that the epoch length $\tau_t$ and the sample sizes $B_t$ and $S_t$ are all adjustable for each epoch $t$. Therefore, besides setting a constant set of parameters, we can also adjust them gradually in order to obtain better theoretical properties and practical performance.

This variance-reduction technique was first proposed as part of SARAH [21] where it is called *recursive* variance reduction. It was also proposed in [8] in the form of a *Stochastic Path-Integrated Differential EstimatoR* (SPIDER). Here we simply call it *incremental* variance reduction. A distinct

feature of this incremental estimator is that the inner-loop estimates $y_i^t$ and $z_i^t$ are biased, i.e.,

$$\begin{cases} \mathbf{E}[y_i^t | x_i^t] = g(x_i^t) - g(x_{i-1}^t) + y_{i-1}^t \neq g(x_i^t), \\ \mathbf{E}[z_i^t | x_i^t] = g'(x_i^t) - g'(x_{i-1}^t) + z_{i-1}^t \neq g'(x_i^t). \end{cases} \tag{16}$$

This is in contrast to two other popular variance-reduction techniques, namely, SVRG [13] and SAGA [6], whose gradient estimators are always unbiased. Note that unbiased estimators for $g(x_i^t)$ and $g'(x_i^t)$ are not essential here, because the composite estimator $\tilde{\nabla} F(x_i^t) = (z_i^t)^T f'(y_i^t)$ is always biased. Therefore the our main task is to control the variance and bias altogether for the proposed estimator.

## 3 Convergence Analysis

In this section, we present theoretical results on the convergence properties of CIVR (Algorithm 1) when the composite function $F$ is smooth. More specifically, we make the following assumptions.

**Assumption 1.** *The following conditions hold concerning problems* (1) *and* (2)*:*

- *$f : \mathbf{R}^p \to \mathbf{R}$ is a $C^1$ smooth and $\ell_f$-Lipschitz function and its gradient $f'$ is $L_f$-Lipschitz.*

- *Each $g_\xi : \mathbf{R}^d \to \mathbf{R}^p$ is a $C^1$ smooth and $\ell_g$-Lipschitz vector mapping and its Jacobian $g_\xi'$ is $L_g$-Lipschtiz. Consequently, $g$ in* (3) *is $\ell_g$-Lipschitz and its Jacobian $g'$ is $L_g$-Lipschitz.*

- *$r : \mathbf{R}^d \to \mathbf{R} \cup \{\infty\}$ is a convex and lower-semicontinuous function.*

- *The overall objective function $\Phi$ is bounded below, i.e., $\Phi^* = \inf_x \Phi(x) > -\infty$.*

**Assumption 2.** *For problem* (1)*, we further assume that there exist constants $\sigma_g$ and $\sigma_{g'}$ such that*

$$\mathbf{E}_\xi[\|g_\xi(x) - g(x)\|^2] \leq \sigma_g^2, \qquad \mathbf{E}_\xi[\|g_\xi'(x) - g'(x)\|^2] \leq \sigma_{g'}^2. \tag{17}$$

As a result of Assumption 1, $F(x) = f\big(g(x)\big)$ is smooth and $F'$ is $L_F$-Lipschitz continuous with

$$L_F = \ell_g^2 L_f + \ell_f L_g$$

(see proof in the supplementary materials). For convenience, we also define two constants

$$G_0 := 2\big(\ell_g^4 L_f^2 + \ell_f^2 L_g^2\big), \qquad \text{and} \qquad \sigma_0^2 := 2\big(\ell_g^2 L_f^2 \sigma_g^2 + \ell_f^2 \sigma_{g'}^2\big). \tag{18}$$

It is important to notice that $G_0 = O(L_F^2)$, hence the step size used later is $\eta = \Theta(1/\sqrt{G_0}) = \Theta(1/L_F)$. We are allowed to use this constant step size mainly due to the assumption that each $g_\xi(\cdot)$ is smooth, instead of the weaker assumption that $E_\xi[g_\xi(x)]$ is smooth as in classical stochastic optimization.

In the next two subsections, we present complexity analysis of CIVR for solving problem (1) and (2) respectively. Due to the space limitation, all proofs are provided in the supplementary materials.

### 3.1 The composite expectation case

The following results for solving problem (1) are presented with notations defined in (3), (14) and (18).

**Theorem 1.** *Suppose Assumptions 1 and 2 hold. Given any $\epsilon > 0$, we set $T = \lceil 1/\sqrt{\epsilon} \rceil$ and*

$$\tau_t = \tau = \lceil 1/\sqrt{\epsilon} \rceil, \quad B_t = B = \lceil \sigma_0^2/\epsilon \rceil, \quad S_t = S = \lceil 1/\sqrt{\epsilon} \rceil, \quad \text{for} \quad t = 1, \dots, T.$$

*Then as long as $\eta \leq \frac{4}{L_F + \sqrt{L_F^2 + 12G_0}}$, the output $\bar{x}$ of Algorithm 1 satisfies*

$$\mathbf{E}\big[\|\mathcal{G}_\eta(\bar{x})\|^2\big] \leq \Big(8\big(\Phi(x_0^1) - \Phi^*\big)\eta^{-1} + 6\Big) \cdot \epsilon = O(\epsilon). \tag{19}$$

*As a result, the sample complexity of obtaining an $\epsilon$-approximate solution is $TB + 2T\tau S = O\big(\epsilon^{-3/2}\big)$.*

Note that in the above scheme, the epoch lengths $\tau_t$ and all the batch sizes $B_t$ and $S_t$ are set to be constant (depending on a pre-fixed $\epsilon$) without regard of $t$. Intuitively, we do not need as many samples in the early stage of the algorithm as in the later stage. In addition, it will be useful in practice to have a variant of the algorithm that can adaptively choose $\tau_t$, $B_t$ and $S_t$ throughout the epochs without dependence on a pre-fixed precision. This is done in the following theorem.

**Theorem 2.** *Suppose Assumptions 1 and 2 hold. We set $\tau_t = S_t = \lceil at + b \rceil$ and $B_t = \lceil \sigma_0^2(at+b)^2 \rceil$ where $a > 0$ and $b \geq 0$. Then as long as $\eta \leq \frac{4}{L_F + \sqrt{L_F^2 + 12G_0}}$, we have for any $T \geq 1$,*

$$\mathbf{E}\left[\|\mathcal{G}_\eta(\bar{x})\|^2\right] \leq \frac{2}{aT^2 + (a+2b)T}\left(\frac{8\left(\Phi(x_0^1) - \Phi^*\right)}{\eta} + \frac{6}{a+b} + \frac{6}{a}\ln\left(\frac{aT+b}{a+b}\right)\right) = O\left(\frac{\ln T}{T^2}\right). \quad (20)$$

*As a result, obtaining an $\epsilon$-approximate solution requires $T = \tilde{O}(1/\sqrt{\epsilon})$ epochs and a total sample complexity of $\tilde{O}(\epsilon^{-3/2})$, where the $\tilde{O}(\cdot)$ notation hides logarithmic factors.*

## 3.2 The composite finite-sum case

In this section, we consider the composite finite-sum optimization problem (2). In this case, the random variable $\xi$ has a uniform distribution over the finite index set $\{1, ..., n\}$. At the beginning of each epoch in Algorithm 1, we use the full sample size $\mathcal{B}_t = \{1, \ldots, n\}$ to compute $y_0^t$ and $z_0^t$. Therefore $B_t = n$ for all $t$ and Equation (8) in Algorithm 1 becomes

$$y_0^t = g(x_0^t) = \frac{1}{n}\sum_{j=1}^n g_j(x_0^t), \qquad z_0^t = g'(x_0^t) = \frac{1}{n}\sum_{j=1}^n g_j'(x_0^t). \quad (21)$$

Also in this case, we no longer need Assumption 2.

**Theorem 3.** *Suppose Assumptions 1 holds. Let the parameters in Algorithm 1 be set as $\mathcal{B}_t = \{1, \ldots, n\}$ and $\tau_t = S_t = \lceil \sqrt{n} \rceil$ for all $t$. Then as long as $\eta \leq \frac{4}{L_F + \sqrt{L_F^2 + 12G_0}}$, we have for any $T \geq 1$,*

$$\mathbf{E}\left[\|\mathcal{G}_\eta(\bar{x})\|^2\right] \leq \frac{8\left(\Phi(x_0^1) - \Phi^*\right)}{\eta\sqrt{n}T} = O\left(\frac{1}{\sqrt{n}T}\right), \quad (22)$$

*As a result, obtaining an $\epsilon$-approximate solution requires $T = O\left(1/(\sqrt{n}\epsilon)\right)$ epochs and a total sample complexity of $TB + 2T\tau S = O\left(n + \sqrt{n}\epsilon^{-1}\right)$.*

Similar to the previous section, we can also choose the epoch lengths and sample sizes adaptively to save the sampling cost in the early stage of the algorithm. However, due to the finite-sum structure of the problem, when the batch size $B_t$ reaches $n$, we will start to take the full batch at the beginning of each epoch to get the exact $g(x_0^t)$ and $g'(x_0^t)$. This leads to the following theorem.

**Theorem 4.** *Suppose Assumptions 1 holds. For some positive constants $a > 0$ and $0 \leq b < \sqrt{n}$, denote $T_0 := \lceil \frac{\sqrt{n}-b}{a} \rceil = O(\sqrt{n})$. When $t \leq T_0$ we set the parameters to be $\tau_t = S_t = \sqrt{B_t} = \lceil at + b \rceil$; when $t > T_0$, we set $\mathcal{B}_t = \{1, \ldots, n\}$ and $\tau_t = S_t = \lceil \sqrt{n} \rceil$. Then as long as $\eta \leq \frac{4}{L_F + \sqrt{L_F^2 + 12G_0}}$,*

$$\mathbf{E}\left[\|\mathcal{G}_\eta(\bar{x})\|^2\right] \leq \begin{cases} O\left(\frac{\ln T}{T^2}\right) & \text{if } T \leq T_0, \\ O\left(\frac{\ln n}{\sqrt{n}(T-T_0+1)}\right) & \text{if } T > T_0. \end{cases} \quad (23)$$

*As a result, the total sample complexity of Algorithm 1 for obtaining an $\epsilon$-approximate solution is $\tilde{O}\left(\min\{\sqrt{n}\epsilon^{-1}, \epsilon^{-3/2}\}\right)$, where $\tilde{O}(\cdot)$ hides logarithmic factors.*

# 4 Fast convergence rates under stronger conditions

In this section we consider two cases where fast linear convergence can be guaranteed for CIVR.

## 4.1 Gradient-dominant function

The first case is when $r \equiv 0$ and $F$ is $\nu$-*gradient dominant*, i.e., there is some $\nu > 0$ such that

$$F(x) - \inf_y F(y) \leq \frac{\nu}{2}\|F'(x)\|^2, \qquad \forall x \in \mathbf{R}^d. \quad (24)$$

Note that a $\mu$-strongly convex function is $(1/\mu)$-gradient dominant by this definition. Hence strong convexity is a special case of the gradient dominant condition, which in turn is a special case of the Polyak-Łojasiewicz condition with the Łojasiewicz exponent equal to 2 [see, e.g., 14].

In order to solve (1) with a pre-fixed precision $\epsilon$, we use a periodic restart strategy as depicted in Algorithm 2. For this restarted version of CIVR, we have the following results.

---
**Algorithm 2:** Restarted CIVR
---
**input:** initial point $\bar{x}^0$, step size $\eta > 0$, number of restarts $K$, number of epochs $T \geq 1$, and a
    set of triples $\{\tau_t, B_t, S_t\}$ for $t = 1, \ldots, T$.
**for** $k = 0, ..., K - 1$ **do**
    |   Generate $\bar{x}^{k+1}$ by Algorithm 1, with parameters $T, \eta, \{\tau_t, B_t, S_t\}$ and initial point $\bar{x}^k$.
**end**
**output:** $\bar{x}^K$.
---

**Theorem 5.** *Consider* (1) *with* $r \equiv 0$. *Suppose Assumptions 1 and 2 hold and F is $\nu$-gradient dominant. For Algorithm 2, given any $\epsilon > 0$, let $\tau_t = S_t = \left\lceil \frac{1}{\sqrt{\epsilon}} \right\rceil$, $B_t = \left\lceil \frac{12\nu\sigma_0^2}{\epsilon} \right\rceil$ and $T = \left\lceil \frac{16\nu\sqrt{\epsilon}}{\eta} \right\rceil$. Then as long as $\eta \leq \frac{4}{L_F + \sqrt{L_F^2 + 12G_0}}$,*

$$\mathbf{E}\left[F(\bar{x}^{k+1}) - F^*\right] \leq \frac{1}{2}\left(F(\bar{x}^k) - F^*\right) + \frac{1}{2}\epsilon. \tag{25}$$

*Consequently, $\mathbf{E}[F(\bar{x}^k) - F^*]$ converges linearly to $\epsilon$ with a factor of $\frac{1}{2}$ per period. The sample complexity for finding an $\epsilon$-solution is $O\left(\left(\nu\epsilon^{-1}\right)\ln \epsilon^{-1}\right)$.*

The restart strategy also applies to the finite-sum case.

**Theorem 6.** *Consider problem* (2) *with* $r \equiv 0$. *Suppose Assumption 1 hold and F is $\nu$-gradient dominant. In Algorithm 2, if we set $\tau_t = S_t = \sqrt{B_t} = \lceil\sqrt{n}\rceil$ and $T = \left\lceil \frac{16\nu}{\sqrt{n}\eta} \right\rceil$, then as long as $\eta \leq \frac{4}{L_F + \sqrt{L_F^2 + 12G_0}}$,*

$$\mathbf{E}\left[F(\bar{x}^{k+1}) - F^*|\bar{x}^k\right] \leq \frac{1}{2}\left(F(\bar{x}^k) - F^*\right). \tag{26}$$

*As a result, the sample complexity for finding an $\epsilon$-solution is $O\left(\left(n + \frac{\nu\sqrt{n}}{\eta}\right)\ln\frac{1}{\epsilon}\right)$.*

It is worth noting that for both cases, the number of epochs $T \propto \eta^{-1}$. When we take more conservative values $\eta$, it will directly result in worse complexity results. This comment also applies to the optimally strongly convex objective function case in the next section.

## 4.2   Optimally strongly convex function

In this part, we assume a *$\mu$-optimally strongly convex* condition on the function $\Phi(x) = F(x) + r(x)$, i.e., there exists a $\mu > 0$ such that

$$\Phi(x) - \Phi(x^*) \geq \frac{\mu}{2}\|x - x^*\|^2, \qquad \forall x \in \mathbf{R}^d. \tag{27}$$

We have the following two results for solving problems (1) and (2) respectively.

**Theorem 7.** *Consider problem* (1). *Suppose Assumptions 1 and 2 hold and $\Phi$ is $\mu$-optimally strongly convex. In Algorithm 2, let us set $\tau_t = S_t = \left\lceil \frac{1}{\sqrt{\epsilon}} \right\rceil$, $B_t = \left\lceil \frac{9\sigma_0^2}{2\mu\epsilon} \right\rceil$ and $T = \lceil \frac{5\sqrt{\epsilon}}{\mu\eta} \rceil$. Then if we choose $\eta < \frac{2}{L_F + \sqrt{L_F^2 + 36G_0}}$,*

$$\mathbf{E}\left[\Phi(\bar{x}^{k+1}) - \Phi^*\right] \leq \frac{1}{2}\left(\Phi(\bar{x}^k) - \Phi^*\right) + \frac{1}{2}\epsilon. \tag{28}$$

*Consquently, $\mathbf{E}[\Phi(\bar{x}^k) - \Phi^*]$ converges linearly to $\epsilon$. The total sample complexity for finding an $\epsilon$-solution is $O\left(\mu^{-1}\epsilon^{-1}\ln \epsilon^{-1}\right)$.*

**Theorem 8.** *Consider the finite-sum problem* (2). *Suppose Assumption 1 hold and $\Phi$ is $\mu$-optimally strongly convex. In Algorithm 2, let us set $\tau_t = S_t = \sqrt{B_t} = \lceil\sqrt{n}\rceil$ and $T = \left\lceil \frac{5}{\sqrt{n}\mu\eta} \right\rceil$. Then if we choose $\eta < \frac{2}{L_F + \sqrt{L_F^2 + 36G_0}}$,*

$$\mathbf{E}\left[\Phi(\bar{x}^{k+1}) - \Phi^*\right] \leq \frac{1}{2}\left(\Phi(\bar{x}^k) - \Phi^*\right). \tag{29}$$

*The sample complexity of finding an $\epsilon$-solution is $O\left(\left(n + \frac{\sqrt{n}}{\mu\eta}\right)\ln\frac{1}{\epsilon}\right)$.*

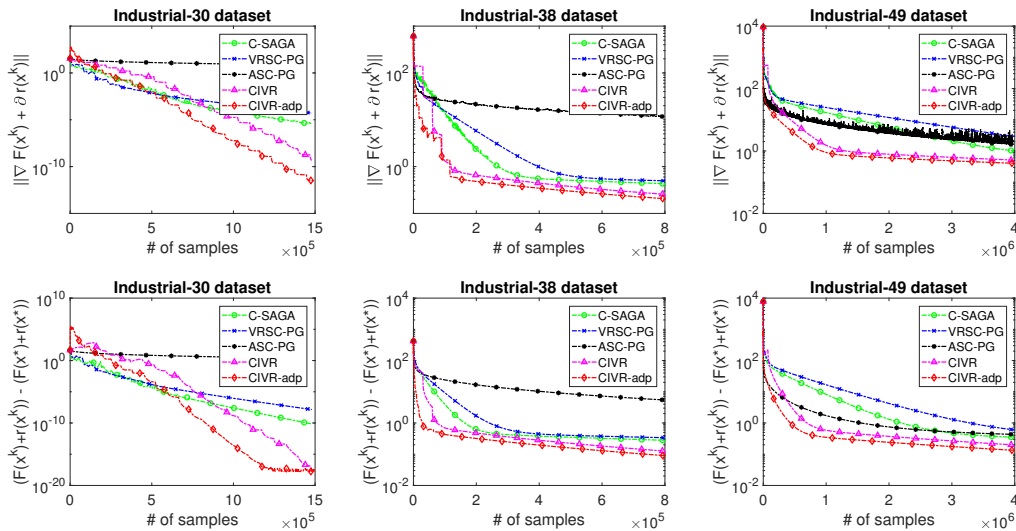

Figure 1: Experiments on the risk-averse portfolio optimization problem.

If we define a condition number $\kappa = L_F/\mu$, then since $\eta = \Theta(1/L_F)$, we have $1/(\mu\eta) = O(\kappa)$ and the above complexities become $O\left(\left(\kappa\epsilon^{-1}\right)\ln\epsilon^{-1}\right)$ and $O\left(\left(n + \kappa n^{1/2}\right)\ln\epsilon^{-1}\right)$.

For Algorithm 2 in both gradient-dominant and strongly convex cases, we have the following remarks.

**Remark 1.** *In Algorithm 2, each run of Algorithm 1 includes a random selection of output. In average it wastes half of the iterates. This waste can be prevented by pre-generating the "stop times" or the output indeces. We can stop Algorithm 1 and output the last iterate whenever the method hits this time.*

**Remark 2.** *In Algorithm 2, the linear-convergence is achieved by restarting. This strategy is proposed partly due to the epoch structure of Algorithm 1. Therefore, if we break this epoch structure by the loopless variance reduction techniques introduced in [16], the restarts may be avoided.*

## 5 Numerical Experiments

In this section, we present numerical experiments for a risk-averse portfolio optimization problem. Suppose there are $d$ assets that one can invest during $n$ time periods labeled as $\{1, ..., n\}$. Let $R_{i,j}$ be the return or payoff per unit of asset $j$ at time $i$, and $R_i$ be the vector consists of $R_{i,1}, \ldots, R_{i,d}$. Let $x \in \mathbf{R}^d$ be the decision variable, where each component $x_j$ represent the amount of investment or percentage of the total investment allocated to asset $j$, for $j = 1, \ldots, d$. The same allocations or percentages of allocations are repeated over the $n$ time periods. We would like to maximize the average return over the $n$ periods, but with a penalty on the variance of the returns across the $n$ periods (in other words, we would like different periods to have similar returns).

This problem is formulated as a mean-variance trade-off:

$$\underset{x\in\mathbf{R}^d}{\text{maximize}} \quad \left\{ \mathbf{E}\big[h_\xi(x)\big] - \lambda\mathbf{Var}\big(h_\xi(x)\big) + r(x) \equiv \mathbf{E}\big[h_\xi(x)\big] - \lambda\Big(\mathbf{E}\big[h_\xi^2(x)\big] - \mathbf{E}\big[h_\xi(x)\big]^2\Big) + r(x)\right\},$$

where the random variable $\xi \in \{1, \ldots, n\}$ takes discrete values uniformly at random and hence makes the problem a finite-sum. The functions $h_i(x) = \langle R_i, x\rangle$ for $i = 1, \ldots, n$ are the rewards. The function $r$ can be chosen as the indicator function of an $\ell_1$ ball, or a soft $\ell_1$ regularization term. We choose the latter one in our experiments to obtain a sparse asset allocation. By using the mappings

$$g_\xi(x) : \mathbf{R}^d \to \mathbf{R}^2 = \big[h_\xi(x)\ h_\xi^2(x)\big]^T, \qquad f(y,z) : \mathbf{R}^2 \to \mathbf{R} = -y + \lambda y^2 - \lambda z,$$

it can be further transformed into the composite finite-sum problem (2), hence readily solved by the CIVR method. Here, the intermediate dimension is very low, i.e., $p = 2$. This leads to very little overhead in computation compared with stochastic optimization without composition.

For comparison, we implement the C-SAGA algorithm [35] as a benchmark. As another benchmark, this problem can also be formulated as a two-layer composite finite-sum problem (7), which was done

in [11] and [19]. We solve the two-layer formulation by ASC-PG [32] and VRSC-PG [11]. Finally, we also implemented CIVR-adp, which is the adaptive sampling variant described in Theorem 4.

We test these algorithms on three real world portfolio datasets, which contain 30, 38 and 49 industrial portfolios respectively, from the Keneth R. French Data Library[1]. For the three datasets, the daily data of the most recent 24452, 10000 and 24400 days are extracted respectively to conduct the experiments. We set the parameter $\lambda = 0.2$ in (5) and use an $\ell_1$ regularization $r(x) = 0.01\|x\|_1$. The experiment results are shown in Figure 1. The curves are averaged over 20 runs and are plotted against the number of samples of the component functions (the horizontal axis).

Throughout the experiments, VRSC-PG and C-SAGA algorithms use the batch size $S = \lceil n^{2/3} \rceil$ while CIVR uses the batch size $S = \lceil \sqrt{n} \rceil$, all dictated by their complexity theory. CIVR-adp employs the adaptive batch size $S_t = \lceil \min\{10t + 1, \sqrt{n}\} \rceil$ for $t = 1, ..., T$. For Industrial-30 dataset, all of VRSC-PG, C-SAGA, CIVR and CIVR-adp use the same step size $\eta = 0.1$. They are chosen from the set $\eta \in \{1, 0.1, 0.01, 0.001, 0.0001\}$ by experiments. And $\eta = 0.1$ works best for all four tested methods simultaneously. Similarly, $\eta = 0.001$ is chosen for the Industrial-38 dataset and $\eta = 0.0001$ is chosen for the Industrial-49 dataset. For ASC-PG, we set its step size parameters $\alpha_k = 0.001/k$ and $\beta_k = 1/k$ [see details in 32]. They are hand-tuned to ensure ASC-PG converges fast among a range of tested parameters. Overall, CIVR and CIVR-adp outperform other methods.

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
