[Supplementary Material]

# A Stochastic Composite Gradient Method with Incremental Variance Reduction

**Junyu Zhang**
University of Minnesota
Minneapolis, Minnesota 55455
zhan4393@umn.edu

**Lin Xiao**
Microsoft Research
Redmond, Washington 98052
lin.xiao@microsoft.com

## Abstract

We consider the problem of minimizing the composition of a smooth (nonconvex) function and a smooth vector mapping, where the inner mapping is in the form of an expectation over some random variable or a finite sum. We propose a stochastic composite gradient method that employs an incremental variance-reduced estimator for both the inner vector mapping and its Jacobian. We show that this method achieves the same orders of complexity as the best known first-order methods for minimizing expected-value and finite-sum nonconvex functions, despite the additional outer composition which renders the composite gradient estimator biased. This finding enables a much broader range of applications in machine learning to benefit from the low complexity of incremental variance-reduction methods.

## 1   Introduction

In this paper, we consider stochastic composite optimization problems

$$\underset{x \in \mathbf{R}^d}{\text{minimize}} \quad f\big(\mathbf{E}_\xi[g_\xi(x)]\big) + r(x), \tag{1}$$

where $f : \mathbf{R}^p \to \mathbf{R}$ is a smooth and possibly nonconvex function, $\xi$ is a random variable, $g_\xi : \mathbf{R}^d \to \mathbf{R}^p$ is a smooth vector mapping for $a.e.$ $\xi$, and $r$ is convex and lower-semicontinuous. A special case we will consider separately is when $\xi$ is a discrete random variable with uniform distribution over $\{1, 2, \ldots, n\}$. In this case the problem is equivalent to a deterministic optimization problem

$$\underset{x \in \mathbf{R}^d}{\text{minimize}} \quad f\left(\frac{1}{n} \sum_{i=1}^{n} g_i(x)\right) + r(x). \tag{2}$$

The formulations in (1) and (2) cover a broader range of applications than classical stochastic optimization and empirical risk minimization (ERM) problems where each $g_\xi$ is a scalar function ($p = 1$) and $f$ is the scalar identity map. Interesting examples include the policy evaluation in reinforcement learning (RL) [e.g., 30], the risk-averse mean-variance optimization ([e.g., 28, 29], through a reformulation by [35]), the stochastic variational inequality ([e.g., 12, 15] through a reformulation in [10]), the 2-level composite risk minimization problems [7], etc.

For the ease of notation, we define

$$g(x) := \mathbf{E}_\xi[g_\xi(x)], \qquad F(x) := f(g(x)), \qquad \Phi(x) := F(x) + r(x). \tag{3}$$

In addition, let $f'$ and $F'$ denote the gradients of $f$ and $F$ respectively, and $g'_\xi(x) \in \mathbf{R}^{p \times d}$ denote the Jacobian matrix of $g_\xi$ at $x$. Then we have

$$F'(x) = \nabla\Big(f\big(\mathbf{E}_\xi[g_\xi(x)]\big)\Big) = \Big(\mathbf{E}_\xi[g'_\xi(x)]\Big)^T f'\big(\mathbf{E}_\xi[g_\xi(x)]\big).$$

Table 1: Sample complexities of CIVR (Composite Incremental Variance Reduction)

| Problem | Assumptions (common: $f$ and $g_\xi$ Lipschitz and smooth, thus $F$ smooth) | | |
| | $F$ nonconvex, $r$ convex | $F$ $\nu$-gradient dominant, $r \equiv 0$ | $F$ convex, $r$ convex, $\Phi$ $\mu$-optimally strongly convex |
| --- | --- | --- | --- |
| (1) | $O\left(\epsilon^{-3/2}\right)$ | $O\left(\left(\nu\epsilon^{-1}\right)\log\epsilon^{-1}\right)$ | $O\left(\left(\mu^{-1}\epsilon^{-1}\right)\log\epsilon^{-1}\right)$ |
| (2) | $O\left(\min\{\epsilon^{-3/2}, n^{1/2}\epsilon^{-1}\}\right)$ | $O\left(\left(n + \nu n^{1/2}\right)\log\epsilon^{-1}\right)$ | $O\left(\left(n + \mu^{-1}n^{1/2}\right)\log\epsilon^{-1}\right)$ |

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

}\left[h_\xi(x)\right] - \lambda\mathbf{Var}\left(h_\xi(x)\right) + r(x) \equiv \mathbf{E}\left[h_\xi(x)\right] - \lambda\left(\mathbf{E}\left[h_\xi^2(x)\right] - \mathbf{E}\left[h_\xi(x)\right]^2\right) + r(x)\right\},$$

where the random variable $\xi \in \{1,\ldots,n\}$ takes discrete values uniformly at random and hence makes the problem a finite-sum. The functions $h_i(x) = \langle R_i, x\rangle$ for $i = 1,\ldots,n$ are the rewards. The function $r$ can be chosen as the indicator function of an $\ell_1$ ball, or a soft $\ell_1$ regularization term. We choose the latter one in our experiments to obtain a sparse asset allocation. By using the mappings

$$g_\xi(x) : \mathbf{R}^d \to \mathbf{R}^2 = \left[h_\xi(x)\ \ h_\xi^2(x)\right]^T, \qquad f(y,z) : \mathbf{R}^2 \to \mathbf{R} = -y + \lambda y^2 - \lambda z,$$

it can be further transformed into the composite finite-sum problem (2), hence readily solved by the CIVR method. Here, the intermediate dimension is very low, i.e., $p = 2$. This leads to very little overhead in computation compared with stochastic optimization without composition.

For comparison, we implement the C-SAGA algorithm [35] as a benchmark. As another benchmark, this problem can also be formulated as a two-layer composite finite-sum problem (7), which was done

in [11] and [19]. We solve the two-layer formulation by ASC-PG [32] and VRSC-PG [11]. Finally, we also implemented CIVR-adp, which is the adaptive sampling variant described in Theorem 4.

We test these algorithms on three real world portfolio datasets, which contain 30, 38 and 49 industrial portfolios respectively, from the Keneth R. French Data Library[1]. For the three datasets, the daily data of the most recent 24452, 10000 and 24400 days are extracted respectively to conduct the experiments. We set the parameter $\lambda = 0.2$ in (5) and use an $\ell_1$ regularization $r(x) = 0.01\|x\|_1$. The experiment results are shown in Figure 1. The curves are averaged over 20 runs and are plotted against the number of samples of the component functions (the horizontal axis).

Throughout the experiments, VRSC-PG and C-SAGA algorithms use the batch size $S = \lceil n^{2/3} \rceil$ while CIVR uses the batch size $S = \lceil \sqrt{n} \rceil$, all dictated by their complexity theory. CIVR-adp employs the adaptive batch size $S_t = \lceil \min\{10t + 1, \sqrt{n}\} \rceil$ for $t = 1, ..., T$. For Industrial-30 dataset, all of VRSC-PG, C-SAGA, CIVR and CIVR-adp use the same step size $\eta = 0.1$. They are chosen from the set $\eta \in \{1, 0.1, 0.01, 0.001, 0.0001\}$ by experiments. And $\eta = 0.1$ works best for all four tested methods simultaneously. Similarly, $\eta = 0.001$ is chosen for the Industrial-38 dataset and $\eta = 0.0001$ is chosen for the Industrial-49 dataset. For ASC-PG, we set its step size parameters $\alpha_k = 0.001/k$ and $\beta_k = 1/k$ [see details in 32]. They are hand-tuned to ensure ASC-PG converges fast among a range of tested parameters. Overall, CIVR and CIVR-adp outperform other methods.

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

# Appendices

## A  Convergence analysis for composite expectation case

In this section, we focus on convergence analysis of CIVR for solving the stochastic composite optimization problem (1), and prove Theorems 1 and 2.

First, we show that under Assumption 1, the composite function $F(x) = f(g(x))$ is smooth and $F'$ has Lipschitz constant $L_f = \ell_g^2 L_f + \ell_f L_g$.

$$
\begin{aligned}
\|F'(x) - F'(y)\| &= \left\| g'(x)^T f'(g(x)) - g'(y)^T f'(g(y)) \right\| \\
&= \left\| g'(x)^T f'(g(x)) - g'(x)^T f'(g(y)) + g'(x)^T f'(g(y)) - g'(y)^T f'(g(y)) \right\| \\
&\leq \left\| g'(x)^T f'(g(x)) - g'(x)^T f'(g(y)) \right\| + \left\| g'(x)^T f'(g(y)) - g'(y)^T f'(g(y)) \right\| \\
&\leq \|g'(x)\| \, \|f'(g(x)) - f'(g(y))\| + \|f'(g(y))\| \, \|g'(x) - g'(y)\| \\
&\leq \|g'(x)\| \cdot L_f \, \|g(x) - g(y)\| + \|f'(g(y))\| \cdot L_g \, \|x - y\| \\
&\leq \ell_g L_f \ell_g \, \|x - y\| + \ell_f L_g \, \|x - y\| \\
&= (\ell_g^2 L_f + \ell_f L_g) \|x - y\|,
\end{aligned}
$$

where we used $\|g'(x)\| \leq \ell_g$ and $\|f'(g(y))\| \leq \ell_f$, which are implied by the Lipschitz conditions on $g$ and $f$ respectively.

Although the incremental estimators used in CIVR are biased, as shown in (16), we can still bound their squared distances from the targets. This is given in the following lemma.

**Lemma 1.** *Suppose Assumption 1 holds. Let $y_i^t$ and $z_i^t$ be constructed according to (8) and (9) in Algorithm 1. For any $t \geq 1$ and $1 \leq i \leq \tau_t - 1$, we have the following mean squared error (MSE) bounds*

$$
\begin{cases}
\mathbf{E}\left[ \|y_i^t - g(x_i^t)\|^2 \right] \leq \mathbf{E}\left[ \|y_0^t - g(x_0^t)\|^2 \right] + \displaystyle\sum_{r=1}^{i} \frac{\ell_g^2}{S_t} \mathbf{E}\left[ \|x_r^t - x_{r-1}^t\|^2 \right], \\[4mm]
\mathbf{E}\left[ \|z_i^t - g'(x_i^t)\|^2 \right] \leq \mathbf{E}\left[ \|z_0^t - g'(x_0^k)\|^2 \right] + \displaystyle\sum_{r=1}^{i} \frac{L_g^2}{S_t} \mathbf{E}\left[ \|x_r^t - x_{r-1}^t\|^2 \right].
\end{cases}
\tag{30}
$$

*Proof.* We first state a fact that allows us to decompose the MSE into a squared bias term and a variance term, that is, for an arbitrary random vector $\zeta$ and a constant vector $u$, we have

$$
\mathbf{E}[\|\zeta - u\|^2] = \|\mathbf{E}[\zeta] - u\|^2 + \mathbf{Var}(\zeta),
\tag{31}
$$

where $\mathbf{Var}(\zeta) := \mathbf{E}[\|\zeta - \mathbf{E}[\zeta]\|^2]$. As a result,

$$
\mathbf{E}\left[ \|y_i^t - g(x_i^t)\|^2 \big| x_i^t \right] = \left\| \mathbf{E}[y_i^t | x_i^t] - g(x_i^t) \right\|^2 + \mathbf{Var}(y_i^t | x_i^t).
$$

For the bias term, we have $\mathbf{E}[y_i^t | x_i^t] - g(x_i^t) = y_{i-1}^t - g(x_{i-1}^t)$. For the variance term, we have

$$
\begin{aligned}
\mathbf{Var}(y_i^t | x_i^t) &= \mathbf{Var}\left( y_{i-1}^t + \frac{1}{S_t} \sum_{\xi \in S_i^t} (g_\xi(x_i^t) - g_\xi(x_{i-1}^t)) \,\Big|\, x_i^t \right) \\
&= \frac{1}{S_t} \mathbf{Var}\left( g_\xi(x_i^t) - g_\xi(x_{i-1}^t) \,|\, x_i^t \right) \\
&\leq \frac{1}{S_t} \mathbf{E}\left[ \|g_\xi(x_i^t) - g_\xi(x_{i-1}^t)\|^2 | x_i^t \right] \\
&\leq \frac{\ell_g^2}{S_t} \|x_i^t - x_{i-1}^t\|^2,
\end{aligned}
$$

where the second equality is due to the fact that $y_{i-1}^t$ is a constant conditioning on $x_i^t$ and in the last inequality we used the $\ell_g$-Lipschitz continuity of $g_\xi$. Consequently,

$$
\mathbf{E}\left[ \|y_i^t - g(x_i^t)\|^2 \right] \leq \mathbf{E}\left[ \|y_{i-1}^t - g(x_{i-1}^t)\|^2 \right] + \frac{\ell_g^2}{S_t} \mathbf{E}\left[ \|x_i^t - x_{i-1}^t\|^2 \right].
$$

Recursively applying the above procedure yields

$$\mathbf{E}\left[\|y_i^t - g(x_i^t)\|^2\right] \leq \mathbf{E}\left[\|y_0^t - g(x_0^t)\|^2\right] + \sum_{r=1}^{i} \frac{\ell_g^2}{S_t} \mathbf{E}\left[\|x_r^t - x_{r-1}^t\|^2\right]. \tag{32}$$

Similarly, the bound on $\mathbf{E}\left[\|z_i^t - g'(x_i^t)\|^2\right]$ can be shown by using the $L_g$-Lipschitz continuity of $g'_\xi$. $\qquad\square$

In Algorithm 1, we approximate the gradient of $F(x) := f(g(x))$ by $\tilde{\nabla}F(x_i^t) = (z_i^t)^T f'(y_i^t)$. The next lemma bounds the MSE of this estimator.

**Lemma 2.** *Suppose Assumptions 1 and 2 hold. Then we have*

$$\mathbf{E}[\|\tilde{\nabla}F(x_i^t) - F'(x_i^t)\|^2] \leq \frac{G_0}{S_t} \sum_{r=1}^{i} \mathbf{E}[\|x_r^t - x_{r-1}^t\|^2] + \frac{\sigma_0^2}{B_t}, \tag{33}$$

*where*

$$G_0 := 2\left(\ell_g^4 L_f^2 + \ell_f^2 L_g^2\right) \quad and \quad \sigma_0^2 := 2\left(\ell_g^2 L_f^2 \sigma_g^2 + \ell_f^2 \sigma_{g'}^2\right).$$

*Proof.* Using Assumption 1, one immediately gets

$$\begin{aligned}
&\mathbf{E}\left[\|\tilde{\nabla}F(x_i^t) - F'(x_i^t)\|^2\right] \\
=\ & \mathbf{E}\left[\|(z_i^t)^T f'(y_i^t) - (g'(x_i^t))^T f'(g(x_i^t))\|^2\right] \\
=\ & \mathbf{E}\left[\|(z_i^t)^T f'(y_i^t) - (g'(x_i^t))^T f'(y_i^t) + (g'(x_i^t))^T f'(y_i^t) - (g'(x_i^t))^T f'(g(x_i^t))\|^2\right] \\
\leq\ & 2\mathbf{E}\left[\|(z_i^t)^T f'(y_i^t) - (g'(x_i^t))^T f'(y_i^t)\|^2\right] + 2\mathbf{E}\left[\|(g'(x_i^t))^T f'(y_i^t) - (g'(x_i^t))^T f'(g(x_i^t))\|^2\right] \\
\leq\ & 2\ell_f^2 \mathbf{E}\left[\|z_i^t - g(x_i^t)\|^2\right] + 2\ell_g^2 L_f^2 \mathbf{E}\left[\|y_i^t - g'(x_i^t)\|^2\right].
\end{aligned} \tag{34}$$

Therefore, by substituting the MSE bounds provided in Lemma 1 into inequality (34), we obtain

$$\begin{aligned}
\mathbf{E}\left[\|\tilde{\nabla}F(x_i^t) - F'(x)\|^2\right] \leq\ & \frac{2\left(\ell_g^4 L_f^2 + \ell_f^2 L_g^2\right)}{S_t} \sum_{r=1}^{i} \mathbf{E}\left[\|x_r^t - x_{r-1}^t\|^2\right] \\
& + 2\ell_g^2 L_f^2 \mathbf{E}\left[\|y_0^t - g(x_0^t)\|^2\right] + 2\ell_f^2 \mathbf{E}\left[\|z_0^t - g'(x_0^k)\|^2\right].
\end{aligned} \tag{35}$$

Under Assumption 2, we can bound the MSE of the estimates in (8) as

$$\mathbf{E}\left[\|y_0^t - g(x_0^t)\|^2\right] \leq \frac{\sigma_g^2}{B_t}, \qquad \mathbf{E}\left[\|z_0^t - g'(x_0^k)\|^2\right] \leq \frac{\sigma_{g'}^2}{B_t}.$$

Combining these MSE bounds with (35) yields the desired result. $\qquad\square$

For the proximal gradient type of algorithms, no matter deterministic or stochastic, a common metric to quantify the optimality of $x_i^t$ is the norm of the so-called *proximal gradient mapping*

$$\mathcal{G}_\eta(x_i^t) := \frac{1}{\eta}\left(x_i^t - \hat{x}_{i+1}^t\right), \tag{36}$$

where $\eta$ is the step size used to produce the update

$$\hat{x}_{i+1}^t = \mathbf{prox}_r^\eta\left(x_i^t - \eta F'(x_i^t)\right).$$

Since we use a constant $\eta$ throughout this paper, we will omit the subscript $\eta$ and use $\mathcal{G}(x)$ to denote the proximal gradient mapping at $x$.

Our goal is to find a point $x$ with $\mathbf{E}\left[\|\mathcal{G}(x)\|^2\right] \leq \epsilon$. However, in Algorithm 1, we only have the approximate proximal gradient mapping

$$\tilde{\mathcal{G}}(x_i^t) := \frac{1}{\eta}(x_i^t - x_{i+1}^t), \tag{37}$$

where $x_{i+1}^t$ is computed using the estimated gradient $\tilde{\nabla}F(x_i^t)$:

$$x_{i+1}^t = \mathbf{prox}_r^\eta\left(x_i^t - \eta\tilde{\nabla}F(x_i^t)\right).$$

Hence we need to establish the connection between $\mathcal{G}(x_i^t)$ and $\tilde{\mathcal{G}}(x_i^t)$, which is done in the next lemma.

**Lemma 3.** *For the two gradient mappings defined in (36) and (37), we have*

$$\mathbf{E}\left[\|\mathcal{G}(x_i^t)\|^2\right] \le 2\mathbf{E}\left[\|\tilde{\mathcal{G}}(x_i^t)\|^2\right] + 2\mathbf{E}\left[\|\tilde{\nabla}F(x_i^t) - F'(x_i^t)\|^2\right]. \tag{38}$$

*Proof.* Using the inequality $\|x_i^t - \hat{x}_{i+1}^t\|^2 \le 2\|x_i^t - x_{i+1}^t\|^2 + 2\|x_{i+1}^t - \hat{x}_{i+1}^t\|^2$ and the definitions of $\mathcal{G}(x_i^t)$ and $\tilde{\mathcal{G}}(x_i^t)$, we have

$$
\begin{aligned}
\mathbf{E}\left[\|\mathcal{G}(x_i^t)\|^2\right] &\le 2\mathbf{E}\left[\|\tilde{\mathcal{G}}(x_i^t)\|^2\right] + \frac{2}{\eta^2}\left\|x_{i+1}^t - \hat{x}_{i+1}^t\right\|^2 \\
&= 2\mathbf{E}\left[\|\tilde{\mathcal{G}}(x_i^t)\|^2\right] + \frac{2}{\eta^2}\left\|\mathbf{prox}_r^\eta\left(x_i^t - \eta F'(x_i^t)\right) - \mathbf{prox}_r^\eta\left(x_i^t - \eta\tilde{\nabla}F(x_i^t)\right)\right\|^2 \\
&\le 2\mathbf{E}\left[\|\tilde{\mathcal{G}}(x_i^t)\|^2\right] + \frac{2}{\eta^2}\left\|x_i^t - \eta F'(x_i^t) - \left(x_i^t - \eta\tilde{\nabla}F(x_i^t)\right)\right\|^2 \\
&= 2\mathbf{E}\left[\|\tilde{\mathcal{G}}(x_i^t)\|^2\right] + 2\mathbf{E}\left[\|\tilde{\nabla}F(x_i^t) - F'(x_i^t)\|^2\right],
\end{aligned}
$$

where in the second inequality we used the non-expansive property of proximal mapping [e.g., 27, Section 31]. □

The next lemma bounds the amount of expected descent per iteration in Algorithm 1.

**Lemma 4.** *Let the sequence $\{x_i^t\}$ be generated by Algorithm 1. Then for all $t \ge 1$ and $0 \le i \le \tau_t - 1$, we have the following two inequalities*

$$\mathbf{E}[\Phi(x_{i+1}^t)] \le \mathbf{E}[\Phi(x_i^t)] - \left(\frac{\eta}{2} - \frac{L_F\eta^2}{2}\right)\mathbf{E}\left[\|\tilde{\mathcal{G}}(x_i^t)\|^2\right] + \frac{\eta}{2}\mathbf{E}[\|\tilde{\nabla}F(x_i^t) - F'(x_i^t)\|^2], \tag{39}$$

*and*

$$
\begin{aligned}
\mathbf{E}[\Phi(x_{i+1}^t)] \le \ &\mathbf{E}[\Phi(x_i^t)] - \frac{\eta}{8}\mathbf{E}[\|\mathcal{G}(x_i^t)\|^2] + \frac{3\eta}{4}\mathbf{E}[\|\tilde{\nabla}F(x_i^t) - F'(x_i^t)\|^2] \\
&- \left(\frac{1}{4\eta} - \frac{L_F}{2}\right)\mathbf{E}\left[\|x_i^t - x_{i+1}^t\|^2\right].
\end{aligned} \tag{40}
$$

*Proof.* By applying the $L_F$-Lipschitz continuity of $F'$ and the optimality of the $\frac{1}{\eta}$-strongly convex subproblem, we have

$$
\begin{aligned}
\Phi(x_{i+1}^t) &= F(x_{i+1}^t) + r(x_{i+1}^t) \\
&\le F(x_i^t) + \langle F'(x_i^t), x_{i+1}^t - x_i^t\rangle + \frac{L_F}{2}\|x_{i+1}^t - x_i^t\|^2 + r(x_{i+1}^t) \\
&= F(x_i^t) + \langle \tilde{\nabla}F(x_i^t), x_{i+1}^t - x_i^t\rangle + \frac{1}{2\eta}\|x_{i+1}^t - x_i^t\|^2 + r(x_{i+1}^t) \\
&\quad + \langle F'(x_i^t) - \tilde{\nabla}F(x_i^t), x_{i+1}^t - x_i^t\rangle - (\frac{1}{2\eta} - \frac{L_F}{2})\|x_{i+1}^t - x_i^t\|^2 \\
&\le F(x_i^t) + r(x_i^t) - \frac{1}{2\eta}\|x_{i+1}^t - x_i^t\|^2 - (\frac{1}{2\eta} - \frac{L_F}{2})\|x_{i+1}^t - x_i^t\|^2 \\
&\quad + \frac{\eta}{2}\|\tilde{\nabla}F(x_i^t) - F'(x_i^t)\|^2 + \frac{1}{2\eta}\|x_{i+1}^t - x_i^t\|^2 \\
&= \Phi(x_i^t) - (\frac{1}{2\eta} - \frac{L_F}{2})\|x_{i+1}^t - x_i^t\|^2 + \frac{\eta}{2}\|\tilde{\nabla}F(x_i^t) - F'(x_i^t)\|^2.
\end{aligned}
$$

Taking the expectation on both sides completes the proof of inequality (39). By inequality (38), we know that

$$-\frac{\eta}{4}\mathbf{E}[\|\tilde{\mathcal{G}}(x_i^t)\|^2] \le -\frac{\eta}{8}\mathbf{E}[\|\mathcal{G}(x_i^t)\|^2] + \frac{\eta}{4}\mathbf{E}[\|\tilde{\nabla}F(x_i^t) - F'(x_i^t)\|^2].$$

Adding this inequality in to (39) yields (40). □

## A.1 Proof of Theorem 1

*Proof.* Because all $\tau_t$, $B_t$ and $S_t$ are taking their values independent of $t$. We denote $\tau = \tau_t$, $B = B_t$ and $S = S_t$ for all $t$ for clarity. By Lemma 4, summing up inequality (40) throughout the $t$-th epoch and applying (33) gives

$$
\begin{aligned}
\frac{\eta}{8} \sum_{i=0}^{\tau-1} \mathbf{E}[\|\mathcal{G}(x_i^t)\|^2] \quad &\leq \quad \mathbf{E}[\Phi(x_0^t)] - \mathbf{E}[\Phi(x_\tau^t)] - \left(\frac{1}{4\eta} - \frac{L_F}{2}\right) \sum_{r=1}^{\tau} \mathbf{E}[\|x_r^t - x_{r-1}^t\|^2] \\
&\quad + \frac{3G_0\eta}{4S} \sum_{i=1}^{\tau-1} \sum_{r=1}^{i} \mathbf{E}[\|x_r^t - x_{r-1}^t\|^2] + \frac{3\sigma_0^2 \eta}{4B} \tau \\
&\leq \quad \mathbf{E}[\Phi(x_0^t)] - \mathbf{E}[\Phi(x_\tau^t)] - \left(\frac{1}{4\eta} - \frac{L_F}{2} - \tau \frac{3G_0\eta}{4S}\right) \sum_{r=1}^{\tau} \mathbf{E}[\|x_r^t - x_{r-1}^t\|^2] \\
&\quad + \frac{3\sigma_0^2 \eta}{4B} \tau,
\end{aligned}
$$

where the second inequality is due to the fact that

$$
\sum_{i=1}^{\tau-1} \sum_{r=1}^{i} \mathbf{E}[\|x_r^t - x_{r-1}^t\|^2] \leq \tau \sum_{r=1}^{\tau} \mathbf{E}[\|x_r^t - x_{r-1}^t\|^2].
$$

When we choose the parameters satisfying $\tau \leq S$, then the coefficient $\frac{1}{4\eta} - \frac{L_F}{2} - \tau \frac{3G_0\eta}{4S} \geq \frac{1}{4\eta} - \frac{L_F}{2} - \frac{3G_0\eta}{4}$ which depends only on the parameter $\eta$ and some constant. If we choose the $\eta$ according to the theorem, then $\frac{1}{4\eta} - \frac{L_F}{2} - \frac{3G_0\eta}{4} \geq 0$, yielding that

$$
\frac{\eta}{8} \sum_{i=0}^{\tau-1} \mathbf{E}[\|\mathcal{G}(x_i^t)\|^2] \leq \mathbf{E}[\Phi(x_0^t)] - \mathbf{E}[\Phi(x_\tau^t)] + \frac{3\sigma_0^2 \eta}{4B} \tau. \tag{41}
$$

Summing this up throughout the epochs gives

$$
\frac{\eta}{8} \sum_{t=1}^{T} \sum_{i=0}^{\tau-1} \mathbf{E}[\|\mathcal{G}(x_i^t)\|^2] \leq \mathbf{E}[\Phi(x_0^1)] - \mathbf{E}[\Phi(x_\tau^T)] + \frac{3\sigma_0^2 \eta}{4B} \tau T \leq \Phi(x_0^1) - \Phi^* + \frac{3\sigma_0^2 \eta}{4B} \tau T,
$$

where we have applied the fact that $x_0^t = x_\tau^{t-1}$. By the random sampling scheme for output $\bar{x}$, we have

$$
\mathbf{E}[\|\mathcal{G}(\bar{x})\|^2] = \frac{1}{\tau T} \sum_{t=1}^{T} \sum_{i=0}^{\tau-1} \mathbf{E}[\|\mathcal{G}(x_i^t)\|^2] \leq \frac{8(\Phi(x_0^1) - \Phi^*)}{\tau T \eta} + \frac{6\sigma_0^2}{B}. \tag{42}
$$

Substitute the values of $T, \tau$ and $B$ gives (19). $\qquad \square$

To simplify presentation, we omit $\lceil \cdot \rceil$ on integer parameters in the following discussion.

- With $\eta \leq \frac{4}{L_F + \sqrt{L_F^2 + 12G_0}}$, and letting $T = 1/\sqrt{\epsilon}$, $B = \sigma_0^2/\epsilon$, and $\tau = S = 1/\sqrt{\epsilon}$, we have

$$
\mathbf{E}[\|\mathcal{G}(\bar{x})\|^2] \leq 8\big((\Phi(x_0^1) - \Phi^*)\eta^{-1} + 6\big)\epsilon,
$$

  and the sample complexity is $T(B + 2\tau S) = O\big(\sigma_0^2 \epsilon^{-3/2} + \epsilon^{-3/2}\big)$, as in our theorem.

- With $\eta \leq \frac{4}{L_F + \sqrt{L_F^2 + 12G_0}}$, and letting $T = 1/\epsilon$, $B = 1 + \sigma_0^2/\epsilon$, and $\tau = S = 1$, we again obtain

$$
\mathbf{E}[\|\mathcal{G}(\bar{x})\|^2] \leq 8\big((\Phi(x_0^1) - \Phi^*)\eta^{-1} + 6\big)\epsilon,
$$

  but the sample complexity is $T(B + 2\tau S) = O\big(\sigma_0^2 \epsilon^{-2} + \epsilon^{-1}\big)$, which is same as in Ghadimi and Lan [9]. For deterministic optimization with $\sigma_0 = 0$, this recovers the $O(\epsilon^{-1})$ complexity.

## A.2 Proof of Theorem 2

*Proof.* Note that for this set of parameters, we still have the relationship that $\tau_t = S_t$. Therefore, within each epoch, (41) is still true with epoch specific $\tau_t$ and $B_t$. Summing this up gives

$$\frac{\eta}{8} \sum_{t=1}^{T} \sum_{i=0}^{\tau_t-1} \mathbf{E}[\|\mathcal{G}(x_i^t)\|^2] \leq \Phi(x_0^1) - \Phi^* + \sum_{t=1}^{T} \frac{3\sigma_0^2 \eta}{4B_t} \tau_t. \tag{43}$$

By the random selection rule of $\bar{x}$, we have

$$\mathbf{E}[\|\mathcal{G}(\bar{x})\|^2] = \frac{1}{\sum_{t=1}^{T} \tau_t} \sum_{t=1}^{T} \sum_{i=0}^{\tau-1} \mathbf{E}[\|\mathcal{G}(x_i^t)\|^2] \leq \frac{8(\Phi(x_0^1) - \Phi^*)}{\eta \sum_{t=1}^{T} \tau_t} + 6\sigma_0^2 \cdot \frac{\sum_{t=1}^{T} \tau_t/B_t}{\sum_{t=1}^{T} \tau_t}. \tag{44}$$

Note that $\tau_t = \lceil at + b \rceil$ and $B_t = \lceil \sigma_0^2 (at + b)^2 \rceil$. We have

$$\sum_{t=1}^{T} \tau_t \geq \sum_{t=1}^{T} at + b = \frac{a}{2} T(T + 1) + bT = O(T^2)$$

and

$$\sigma_0^2 \sum_{t=1}^{T} \tau_t/B_t \leq \sum_{t=1}^{T} \frac{1}{at + b} \leq \frac{1}{a + b} + \int_1^T \frac{dt}{at + b} = \frac{1}{a + b} + \frac{1}{a} \ln\left(\frac{aT + b}{a + b}\right) = O(\ln T).$$

Substituting the above bounds into inequality (44) gives (20). As a result, the total sample complexity is

$$\sum_{t=1}^{T} (B_t + 2\tau_t S_t) \leq \sum_{t=1}^{T} \left(\sigma_0^2 (at + b)^2 + 2(at + b)^2\right) = O(\sigma_0^2 T^3 + T^3).$$

Setting $T = \tilde{O}(\epsilon^{-1/2})$ so that $\mathbf{E}[\|\mathcal{G}\|\bar{x}\|^2] \leq \epsilon$, we get sample complexity $\tilde{O}(\sigma_0^2 \epsilon^{-3/2} + \epsilon^{-3/2})$. □

We can also choose a different set of parameters. With $\eta \leq \frac{4}{L_F + \sqrt{L_F^2 + 12G_0}}$, and letting $B = 1 + \sigma_0^2 (at + b)$, and $\tau = S = 1$, we also have

$$\mathbf{E}[\|\mathcal{G}(\bar{x})\|^2] \leq \frac{8(\Phi(x_0^1) - \Phi^*)}{\eta T} + \frac{6 \ln T}{T},$$

but the sample complexity, by setting $T = \tilde{O}(\epsilon^{-1})$ so that the above bound is less than $\epsilon$, is

$$\sum_{t=1}^{T} (B_t + 2\tau_t S_t) \leq \sum_{t=1}^{T} \left(\sigma_0^2 (at + b) + 2\right) = O(\sigma_0^2 T^2 + T) = \tilde{O}(\sigma_0^2 \epsilon^{-2} + \epsilon^{-1}).$$

This is more close to the classical results on stochastic optimization.

# B Convergence analysis for composite finite-sum case

In this section, we consider the composite finite-sum problem (2) and prove Theorems 3 and 4.

In this case, the random variable $\xi$ uniformly takes value from the finite index set $\{1, ..., n\}$. At the beginning of each epoch in Algorithm 1, we can choose to estimate $g(x_0^t)^t$ and $g'(x_0^t)$ by their exact value rather than the approximate ones constructed by subsampling. Namely, in (8) of Algorithm 1, we choose $\mathcal{B}_t = \{1, \ldots, n\}$ for all $t \geq 1$. Therefore,

$$y_0^t = g(x_0^t) = \frac{1}{n} \sum_{j=1}^{n} g_j(x_0^t), \qquad z_0^t = g'(x_0^t) = \frac{1}{n} \sum_{j=1}^{n} g_j'(x_0^t)$$

and

$$\mathbf{E}\left[\|y_0^t - g(x_0^t)\|^2\right] = 0, \qquad \mathbf{E}\left[\|z_0^t - g'(x_0^k)\|^2\right] = 0. \tag{45}$$

As a result, the initial variances in Lemma 1 diminishes and (30) reduces to

$$\begin{cases} \mathbf{E}[\|y_i^t - g(x_i^t)\|^2] \leq \sum_{r=1}^{i} \frac{\ell_g^2}{S_t} \mathbf{E}[\|x_r^t - x_{r-1}^t\|^2], \\ \mathbf{E}[\|z_i^t - g'(x_i^t)\|^2] \leq \sum_{r=1}^{i} \frac{L_g^2}{S_t} \mathbf{E}[\|x_r^t - x_{r-1}^t\|^2]. \end{cases} \tag{46}$$

In addition, combining (35) and (45), we have

$$\mathbf{E}[\|\tilde{\nabla}F(x_i^t) - F'(x)\|^2] \leq \frac{G_0}{S_t} \sum_{r=1}^{i} \mathbf{E}[\|x_r^t - x_{r-1}^t\|^2]. \tag{47}$$

Note that Lemma 4 is still true.

## B.1 Proof of Theorem 3

*Proof.* The proof follows similar steps as those in the proof of Theorem 1. So we only note down the significantly different steps here.

Specifically, following the proof of Theorem 1 in Section A.1, by applying (46) instead of (30), we get the following result instead of inequality (41),

$$\frac{\eta}{8} \sum_{i=0}^{\tau-1} \mathbf{E}[\|\mathcal{G}(x_i^t)\|^2] \leq \mathbf{E}[\Phi(x_0^t)] - \mathbf{E}[\Phi(x_\tau^t)].$$

Summing this up apply the random selection rule of $\bar{x}$ gives

$$\mathbf{E}[\|\mathcal{G}(\bar{x})\|^2] = \frac{1}{\tau T} \sum_{t=1}^{T} \sum_{i=0}^{\tau-1} \mathbf{E}[\|\mathcal{G}(x_i^t)\|^2] \leq \frac{8(\Phi(x_0^1) - \Phi^*)}{\tau T \eta} = \frac{8(\Phi(x_0^1) - \Phi^*)}{\sqrt{n} T \eta}.$$

Therefore, we have to set $T = O(\frac{1}{\sqrt{n}\epsilon})$ to get an $\epsilon$-solution. Note that the sample complexity per epoch is $n + \tau_t S_t = 2n$, the total sample complexity will be $O(n + \sqrt{n}\epsilon^{-1})$. $\square$

## B.2 Proof of Theorem 4

*Proof.* If $T \leq T_0$, then the result is exactly what we proved from Theorem 2. Therefore, the first bound in (23) is already guaranteed.

If $T > T_0$, when $1 \leq t \leq T_0$, then everything still runs identically to that described in Theorem 2. Consequently, the following bound is effective

$$\frac{\eta}{8} \sum_{t=1}^{T_0} \sum_{i=0}^{\tau_t-1} \mathbf{E}[\|\mathcal{G}(x_i^t)\|^2] \leq \Phi(x_0^1) - \mathbf{E}[\Phi(x_0^{T_0+1})] + \sum_{t=1}^{T_0} \frac{3\sigma_0^2 \eta}{4B_t} \tau_t. \tag{48}$$

When $T_0 + 1 \leq t \leq T$, the following bound becomes effective,

$$\frac{\eta}{8} \sum_{t=T_0+1}^{T} \sum_{i=0}^{\tau-1} \mathbf{E}[\|\mathcal{G}(x_i^t)\|^2] \leq \mathbf{E}[\Phi(x_0^{T_0+1})] - \Phi^*.$$

Therefore, we have

$$\mathbf{E}[\|\mathcal{G}(\bar{x})\|^2] = \frac{1}{\sum_{t=1}^{T} \tau_t} \sum_{t=1}^{T} \sum_{i=0}^{\tau-1} \mathbf{E}[\|\mathcal{G}(x_i^t)\|^2] \leq \frac{8(\Phi(x_0^1) - \Phi^*)}{\eta \sum_{t=1}^{T} \tau_t} + 6\sigma_0^2 \cdot \frac{\sum_{t=1}^{T_0} \tau_t/B_t}{\sum_{t=1}^{T} \tau_t}.$$

Note that

$$\sum_{t=1}^{T_0} \tau_t/B_t \leq \sum_{t=1}^{T_0} \frac{1}{at+b} \leq \frac{1}{a+b} + \frac{1}{a} \ln\left(\frac{aT_0+b}{a+b}\right) = O(\ln n),$$

and

$$\sum_{t=1}^{T} \tau_t \geq (T - T_0)\sqrt{n} + \sum_{t=1}^{T_0} (at+b) = \sqrt{n}(T - T_0) + \frac{a}{2}T_0^2 + (\frac{a}{2} + b)T_0 = O(\sqrt{n}(T - T_0 + 1)).$$

With the above two bounds, we have proved the second result in (23).

For any $\epsilon > 0$, if $\epsilon \geq O(1/T_0^2) = O(n^{-1})$. In this case, the algorithm will spend most epochs in the adaptive phase, whose sample complexity is $\tilde{O}(\epsilon^{-3/2})$. if $\epsilon = o(n^{-1})$, we need $T > T_0$. By (23), we know $\sqrt{n}(T - T_0 + 1) = \tilde{O}(\epsilon^{-1})$, this means that the total sample complexity will be

$$\sum_{t=1}^{T} (B_t + 2\tau_t S_t) \leq 3\sum_{t=1}^{T_0} (at+b+1)^2 + 3(T - T_0)n = \tilde{O}(n^{3/2} + \sqrt{n}\epsilon^{-1}) = \tilde{O}(\sqrt{n}\epsilon^{-1}).$$

When $\epsilon \geq O(n^{-1})$, we have $\epsilon^{-3/2} \leq \sqrt{n}\epsilon^{-1}$. When $\epsilon = o(n^{-1})$, we have $\epsilon^{-3/2} > \sqrt{n}\epsilon^{-1}$. Combining the two cases together gives the sample complexity of $\tilde{O}(\min\{\sqrt{n}\epsilon^{-1}, \epsilon^{-3/2}\})$. $\square$

# C Convergence analysis under gradient-dominant condition

## C.1 Proof of Theorem 5

*Proof.* For the ease of notation, let us only focus in one run of Algorithm 1. And denote the input point as $x_0^1$ and the output point $\bar{x}$. Note that in this case $\Phi(x) = F(x)$. By (42) and (24), we have

$$\mathbf{E}[F(\bar{x}) - F^*] \le \nu \mathbf{E}[\|F'(\bar{x})\|^2] \le \frac{8\nu(F(x_0^1) - F^*)}{\tau T \eta} + \frac{6\nu\sigma_0^2}{B}$$

By the selection of $T = \lceil \frac{16\nu\sqrt{\epsilon}}{\eta} \rceil$, $\tau = 1/\sqrt{\epsilon}$ and $B = 1 + 12\nu\sigma_0^2/\epsilon$, we have

$$\mathbf{E}[F(\bar{x}) - F^*] \le \frac{1}{2}(F(x_0^1) - F^*) + \frac{1}{2}\epsilon, \tag{49}$$

which is (25).

Suppose we periodically restart the Algorithm 1 after every $T$ epochs, and set the outputs to be $\bar{x}^k$, where $k = 1, 2, ...$ denotes the number of restarts. We use the output of the $k$th period $\bar{x}^k$ as the initial point to start the next period, which produces $\bar{x}^{k+1}$. As a result, the above inequality translates to

$$\mathbf{E}[F(\bar{x}^{k+1}) - F^*] \le \frac{1}{2}(\mathbf{E}[F(\bar{x}^k)] - F^*) + \frac{1}{2}\epsilon.$$

Equivalently,

$$\mathbf{E}[F(\bar{x}^k) - F^*] - \epsilon \le \frac{1}{2}\left(\mathbf{E}[F(\bar{x}^{k-1}) - F^*] - \epsilon\right),$$

which leads to

$$\mathbf{E}[F(\bar{x}^k) - F^*] \le \frac{1}{2^k}\left(\mathbf{E}[F(\bar{x}^0) - F^*] - \epsilon\right) + \epsilon.$$

Therefore, the expected optimality gap converges linearly to a $\epsilon$-ball around 0. $\qquad\square$

Next we discuss the sample complexity with different parameter settings.

- If we choose $\tau = S = 1/\sqrt{\epsilon}$, $B_t = 12\nu\sigma_0^2/\epsilon$, and $T = \lceil \frac{16\nu\sqrt{\epsilon}}{\eta} \rceil$, then the total sample complexity is

$$T(B + 2\tau S)\ln\frac{1}{\epsilon} = \frac{16\nu\sqrt{\epsilon}}{\eta}\left(\frac{12\nu\sigma_0^2}{\epsilon} + \frac{1}{\sqrt{\epsilon}}\frac{1}{\sqrt{\epsilon}}\right)\ln\frac{1}{\epsilon} = O\left((\nu^2\sigma_0^2\epsilon^{-1/2} + \nu\epsilon^{-1/2})\ln\epsilon^{-1}\right)$$

  However, the above derivation needs to assume $\frac{16\nu\sqrt{\epsilon}}{\eta} \ge 1$ or at least $O(1)$, which means $\epsilon > (\eta/\nu)^2$. If this condition is not satisfied, then we have $T = 1$ and the complexity is

$$O\left((\nu\sigma_0^2\epsilon^{-1} + \epsilon^{-1})\ln\epsilon^{-1}\right).$$

  Notice that the second term does not depend on $\nu$ or the conditions number.

- If we choose $\tau = S = 1$, $B_t = 1 + 12\nu\sigma_0^2/\epsilon$, and $T = \lceil \frac{16\nu}{\eta} \rceil$, the we also have

$$\mathbf{E}[F(\bar{x}) - F^*] \le \frac{1}{2}(F(x_0^1) - F^*) + \frac{1}{2}\epsilon,$$

  and the total sample complexity is

$$T(B + 2\tau S)\ln\frac{1}{\epsilon} = \frac{16\nu}{\eta}\left(\frac{12\nu\sigma_0^2}{\epsilon} + 2\right)\ln\frac{1}{\epsilon} = O\left(\nu^2\sigma_0^2\epsilon^{-1/2} + \nu\right)\ln\epsilon^{-1}$$

  Defining the condition number $\kappa = L_F\nu = O(\nu/\eta)$, the above complexity becomes

$$T(B + 2\tau S)\ln\frac{1}{\epsilon} = O\left(\kappa^2\sigma_0^2\epsilon^{-1} + \kappa\right)\ln\epsilon^{-1}$$

  Thus when $\sigma = 0$, we have $O(\kappa\ln\epsilon^{-1})$ for deterministic optimization.

## C.2 Proof of Theorem 6

The proof is very similar to the previous one. It actually becomes simpler by noticing that in the finite-sum case, the terms involving $\sigma_0^2$ disappear:

$$\mathbf{E}[F(\bar{x}) - F^*] \leq \nu \mathbf{E}[\|F'(\bar{x})\|^2] \leq \frac{8\nu(F(x_0^1) - F^*)}{\tau T \eta}.$$

By choosing $T = \lceil \frac{16\nu}{\eta \sqrt{n}} \rceil$, $\tau = S = \sqrt{n}$. we again obtain (49). In this case, we have $B = n$ and

$$T(B + 2\tau S)\epsilon^{-1} = \left\lceil \frac{16\nu}{\eta \sqrt{n}} \right\rceil \left( n + 2\sqrt{n}\sqrt{n} \right) \ln \epsilon^{-1} = O\left( n + \nu \sqrt{n} \right) \ln \epsilon^{-1}.$$

# D    Convergence analysis under optimally strong convexity

In order to prove Theorems 7 and 8, we first state Lemma 3 in [34] in our notations.

**Lemma 5** (Lemma 3 in [34]). *Let $\Phi(x) = F(x) + r(x)$, where $F'(x)$ is $L_F$-Lipschitz continuous, and $F(x)$ and $r(x)$ are convex. For any $x \in \text{dom}(r)$, and any $v \in \mathbf{R}^d$, define*

$$x^+ := \text{Prox}_{\eta r(\cdot)}(x - \eta v), \; \mathcal{G} := \frac{1}{\eta}(x - x^+), \; and \; \Delta := v - F'(x),$$

*where $\eta$ is a step size satisfying $0 < \eta \leq 1/L_F$. Then for any $y \in \mathbf{R}^d$,*

$$\Phi(y) \geq \Phi(x^+) + \mathcal{G}^T(y - x) + \frac{\eta}{2}\|\mathcal{G}\|^2 + \Delta^T(x^+ - y).$$

## D.1    Proof of Theorem 7

*Proof.* For the ease of notation, let us only focus in one run of Algorithm 1. And denote the input point as $x_0^1$ and the output point $\bar{x}$. If we set $x = x_i^t$, $y = x^*$, $v = \tilde{\nabla}F(x_i^t)$, $x^+ = x_{i+1}^t$ and $\mathcal{G} = \tilde{\mathcal{G}}(x_i^t)$, we get the following useful inequality,

$$\langle \tilde{\mathcal{G}}(x_i^t), x^* - x_i^t \rangle \leq \Phi(x^*) - \Phi(x_{i+1}^t) - \frac{\eta}{2}\|\tilde{\mathcal{G}}(x_i^t)\|^2 - \langle F'(x_i^t) - \tilde{\nabla}F(x_i^t), x^* - x_{i+1}^t \rangle.$$

As a result we have the following inequality,

$$
\begin{aligned}
& \|x_{i+1}^t - x^*\|^2 \\
= \; & \|x_i^t - x^*\|^2 + \eta^2\|\tilde{\mathcal{G}}(x_i^t)\|^2 + 2\eta\langle \tilde{\mathcal{G}}(x_i^t), x^* - x_i^t \rangle \\
\leq \; & \|x_i^t - x^*\|^2 + \eta^2\|\tilde{\mathcal{G}}(x_i^t)\|^2 - 2\eta(\Phi(x_{i+1}^t) - \Phi(x^*)) - \eta^2\|\tilde{\mathcal{G}}(x_i^t)\|^2 \\
& -2\eta\langle F'(x_i^t) - \tilde{\nabla}F(x_i^t), x^* - x_{i+1}^t \rangle \\
\leq \; & \|x_i^t - x^*\|^2 - 2\eta(\Phi(x_{i+1}^t) - \Phi(x^*)) + \frac{2\eta}{\mu}\|F'(x_i^t) - \tilde{\nabla}F(x_i^t)\|^2 + \frac{\eta\mu}{2}\|x_{i+1}^t - x^*\|^2 \\
\leq \; & \|x_i^t - x^*\|^2 - \eta(\Phi(x_{i+1}^t) - \Phi(x^*)) + \frac{2\eta}{\mu}\|F'(x_i^t) - \tilde{\nabla}F(x_i^t)\|^2. \quad (50)
\end{aligned}
$$

Note that the inequality (50) is originally obtained in [35]. Adding $2\mu \cdot$(50) to (39), we get

$$
\begin{aligned}
2\mu\eta\mathbf{E}[\Phi(x_{i+1}^t) - \Phi^*] \; \leq \; & \mathbf{E}[\Phi(x_i^t) + 2\mu\|x_i^t - x^*\|^2] - \mathbf{E}[\Phi(x_{i+1}^t) + 2\mu\|x_{i+1}^t - x^*\|^2] \\
& -(\frac{1}{2\eta} - \frac{L_F}{2})\mathbf{E}[\|x_{i+1}^t - x_i^t\|^2] + \frac{9}{2}\eta\mathbf{E}[\|\tilde{\nabla}F(x_i^t) - F'(x_i^t)\|^2]. \quad (51)
\end{aligned}
$$

By (51) and (33), we have

$$
\begin{aligned}
2\mu\eta \sum_{i=0}^{\tau_t - 1} \mathbf{E}[\Phi(x_{i+1}^t) - \Phi^*] \; \leq \; & \mathbf{E}[\Phi(x_{\tau_t}^t) + 2\mu\|x_{\tau_t}^t - x^*\|^2] - \mathbf{E}[\Phi(x_0^t) + 2\mu\|x_0^t - x^*\|^2] \\
& -(\frac{1}{2\eta} - \frac{L_F}{2} - \tau_t \frac{9G_0\eta}{2S_t})\sum_{r=1}^{\tau_t} \mathbf{E}[\|x_r^t - x_{r-1}^t\|^2] + \tau_t \frac{9\sigma_0^2\eta}{2B_t}.
\end{aligned}
$$

According to the selection of $\tau_t, S_t, B_t$ and $\eta$, we know that the coefficient $(\frac{1}{2\eta} - \frac{L_F}{2} - \tau_t \frac{9G_0\eta}{2S_t}) \geq 0$. Consequently,

$$2\mu\eta \sum_{i=0}^{\tau_t-1} \mathbf{E}[\Phi(x_{i+1}^t) - \Phi^*] \quad \leq \quad \mathbf{E}[\Phi(x_{\tau_t}^t) + 2\mu\|x_{\tau_t}^t - x^*\|^2] - \mathbf{E}[\Phi(x_0^t) + 2\mu\|x_0^t - x^*\|^2] + \tau_t \frac{9\sigma_0^2\eta}{2B_t}.$$

Summing this up and apply the random selection rule of $\bar{x}$ gives

$$\mathbf{E}[\Phi(\bar{x}) - \Phi^*] \quad \leq \quad \frac{1}{2\mu\eta\tau T}\mathbf{E}[\Phi(x_0^1) - \Phi^* + 2\mu\|x_0^1 - x^*\|^2] + \frac{9\sigma_0^2}{4\mu B_t}$$

$$\leq \quad \frac{5}{2\mu\eta\tau T}\mathbf{E}[\Phi(x_0^1) - \Phi^*] + \frac{9\sigma_0^2}{4\mu B_t}.$$

If we choose $T = \lceil\frac{5\sqrt{\epsilon}}{\mu\eta}\rceil$, $\tau = S = \frac{1}{\sqrt{\epsilon}}$ and $B_t = 1 + \frac{9\sigma_0^2}{2\mu\epsilon}$, then $\frac{5}{2\mu\eta\tau T} \leq \frac{1}{2}$ and we obtain

$$\mathbf{E}[\Phi(\bar{x}) - \Phi^*] \quad \leq \quad \frac{1}{2}\mathbf{E}[\Phi(x_0^1) - \Phi^*] + \frac{1}{2}\epsilon.$$

This proves the inequality (28). The rest of the proof will mimic that of Theorem 5. $\square$

Discussions on sample complexity:

- If we choose $\tau = S = 1/\sqrt{\epsilon}$, $B_t = 1 + \frac{9\sigma_0^2}{2\mu\epsilon}$, and $T = \lceil\frac{5\sqrt{\epsilon}}{\mu\eta}\rceil$, then the sample complexity is

$$T(B + 2\tau S)\ln\frac{1}{\epsilon} = \frac{5\sqrt{\epsilon}}{\mu\eta}\left(\frac{9\sigma_0^2}{2\mu\epsilon} + \frac{1}{\sqrt{\epsilon}}\frac{1}{\sqrt{\epsilon}}\right)\ln\frac{1}{\epsilon} = O\left((\mu^{-2}\sigma_0^2\epsilon^{-1/2} + \mu^{-1}\epsilon^{-1/2})\ln\epsilon^{-1}\right).$$

  The above derivation needs to assume $\frac{5\sqrt{\epsilon}}{\mu\eta} \geq 1$ or at least $O(1)$, which means $\epsilon > (\eta\mu)^2$. If this condition is not satisfied, then we have $T = 1$ and the complexity is

$$O\left((\mu^{-1}\sigma_0^2\epsilon^{-1} + \epsilon^{-1})\ln\epsilon^{-1}\right).$$

- If we choose $\tau = S = 1$, $B_t = 1 + \frac{9\sigma_0^2}{\mu\epsilon}$, and $T = \lceil\frac{5}{\mu\eta}\rceil$, the we also have

$$\mathbf{E}[F(\bar{x}) - F^*] \leq \frac{1}{2}(F(x_0^1) - F^*) + \frac{1}{2}\epsilon,$$

  and the total sample complexity is

$$T(B + 2\tau S)\ln\frac{1}{\epsilon} = \frac{5}{\mu\eta}\left(\frac{9\sigma_0^2}{\mu\epsilon} + 2\right)\ln\frac{1}{\epsilon} = O\left(\mu^{-2}\sigma_0^2\epsilon^{-1} + \mu^{-1}\right)\ln\epsilon^{-1}$$

  Defining the condition number $\kappa = L_F v = O(1/(\mu\eta))$, the above complexity becomes

$$T(B + 2\tau S)\ln\frac{1}{\epsilon} = O\left(\kappa^2\sigma_0^2\epsilon^{-1} + \kappa\right)\ln\epsilon^{-1}$$

  Thus when $\sigma = 0$, we have $O\left(\kappa\ln\epsilon^{-1}\right)$ for deterministic optimization.

## D.2 Proof of Theorem 8

The proof is very similar to the previous one. It actually becomes simpler by noticing that in the finite-sum case, the terms involving $\sigma_0^2$ disappear.

Figure 2: Experiments on policy evaluation for MDP for cases with $S = 10$, $S = 100$ and $S = 500$.

# E   Numerical experiments on policy evaluation for MDP

Here we provide additional numerical experiments on the policy evaluation problem for MDP.

Let $\mathcal{S} = \{1, ..., S\}$ be the state space of some Markov decision process. Suppose a reward of $R_{i,j}$ is received after transitioning from state $i$ to state $j$. Let $P^\pi \in \mathbf{R}^{S \times S}$ be the transition probability matrix under some fixed policy $\pi$. Then the evaluation of the value function $V^\pi : \mathcal{S} \to \mathbf{R}$ under such policy is equivalent to solving the following Bellman equation:

$$V^\pi(i) = \sum_{j=1}^{S} P_{i,j}^\pi (R_{i,j} + \gamma V^\pi(j)) = \mathbf{E}_{j|i}[R_{i,j} + \gamma V^\pi(j)].$$

Following the suggestion of [5, 32], we apply the linear function approximation $V^\pi(i) \approx \langle \Psi_i, w^* \rangle$ for a given set of feature vectors $\Psi_i$. and would like to compute the optimal vector $w^*$. This can be formulated as the following problem

$$\underset{w}{\text{minimize}} \ \ F(w) \triangleq \sum_{i=1}^{S} \left( \langle \Psi_i, w \rangle - \sum_{j=1}^{S} P_{i,j}^\pi (R_{i,j} + \gamma \langle \Psi_j, w \rangle) \right)^2.$$

Let's denote

$$q_i^\pi(w) \triangleq \sum_{j=1}^{S} P_{i,j}^\pi (R_{i,j} + \gamma \langle \Psi_j, w \rangle) = \mathbf{E}_{j|i}[R_{i,j} + \gamma \langle \Psi_j, w \rangle].$$

Then by defining

$$g(w) = \left[ \langle \Psi_1, w \rangle, ..., \langle \Psi_S, w \rangle, q_i^\pi(w), ..., q_S^\pi(w) \right]^T$$

and

$$f(y_1, ..., y_S, z_1, ..., z_S) = \|y - z\|^2 = \sum_{i=1}^{S} (y_i - z_i)^2,$$

the Least squares problem is transformed into the form of (2).

For this problem, we test the SCGD [31], the ASCGD [31], the ASC-PG [32], the VRSC-PG [11], C-SAGA [35] and our CIVR algorithms. In Section 5, we already tested the algorithms under their standard batch sizes, e.g. $\lceil n^{2/3} \rceil$ and $\lceil \sqrt{n} \rceil$. However, small constant batch sizes are often preferred in practice. Therefore, we would like to set the batch size to $s = 1$ for all algorithms. For this special case, we denote the CIVR as the CIVR-b1. To balance the sample complexity between the initial full batch sampling and the later subsampling with $s = 1$, we set the epoch length for VRSC-PG and CIVR-b1 to be $S$.

Note that the last $S$ components of $g$ are all independent expectations, therefore the variance reduction technique of VRSC-PG [11], C-SAGA [35] and CIVR-b1 applied to each of these components. In the experiments, $P^{\pi}$, $\Phi$ and $R^{\pi}$ are generated randomly.

Similar to the experiments performed in Section 5, the step sizes are chosen from $\{0.1, 0.05, 0.01, 0.005, 0.001, 0.0005, 0.0001\}$ by experiments for VRSC-PG, C-SAGA as well as for CIVR-b1. For $S = 10$, $\eta = 0.1$ works best for both C-SAGA and CIVR-b1, while $\eta = 0.01$ works best for VRSC-PG; For $S = 100$, $\eta = 0.001$ works best for both C-SAGA and CIVR-b1, while $\eta = 0.0001$ works best for VRSC-PG. For $S = 500$, $\eta = 0.0001$ works best for all three of them.

When $S = 10$ and $S = 100$, we choose $\alpha_k = 0.01k^{-3/4}$ and $\beta_k = 0.1k^{-1/2}$ for SCGD, $\alpha_k = 0.01k^{-5/7}$ and $\beta_k = 0.1k^{-4/7}$ for ASCGD and $\alpha_k = 0.01k^{-1/2}$ and $\beta_k = 0.1k^{-1}$ for ASC-PG. When $S = 500$, we choose $\alpha_k = 0.0001k^{-3/4}$ and $\beta_k = 0.001k^{-1/2}$ for SCGD while ASCGD and ASC-PG fail to converge under various trials of parameters. The meaning of these step size parameters can be found in [32] and [31].

Figure 2 shows three experiments with sizes $S = 10$, $S = 100$ and $S = 500$ respectively. We can see that both C-SAGA and CIVR-b1 preform much better than other algorithms in our setting. CIVR-b1 has more smooth and stable trajectory than C-SAGA.