[Reviews · NeurIPS 2019]

Reviewer 1



The paper proposes a variance reduction optimization method for stochastic composite optimization problem where the objective function is the composition of a smooth function and an expectation over some random variables. In particular, this problem formulation covers the standard empirical risk minimization and other more advanced target function like risk-averse/robust optimization involving the variance. This problem is very challenging because unbiased gradient estimator is no longer accessible due to the composition of expectation. Therefore one needs to carefully use the biased gradient. The proposed algorithm is based on recursive variance reduction technique in the spirit of SARAH/SPIDER. Extensive convergence analysis are provided under different scenarios. Overall, the paper is very clear and easy to follow, I am quite positive of the paper even though I have some minor concerns. Here are my detailed comments. The main contribution of the paper is to carefully address the biased gradient issue. In the current presentation, it is unclear (at least in the main text) how this is handled. After a quick check on the supplementary material, it seems to me that the way to handle it is to perform an inexact stochastic methods where we are suffering from some additive noise on the gradients. If this is the case, then we could apply standard analysis with inexact gradients. It is possible that I have missed some intrinsic difficulty, please comment on it and provide some intuition how the biased gradient estimator is handled. In typical non-convex optimization, the step size of the algorithm needs to be small , which usually depends on \epsilon, like in SPIDER. It does not seem to be the case here, please comment on it. ===Edit after rebuttal === I thank the authors for clarifying my concerns and I raise my score by 1 accordingly.

Reviewer 2



UPDATE: I strongly urge the authors to 'revise the introduction to be more concise and rely on reference to [34].' as promised in the rebuttal. And also to include a bit more prominently 'While [34] first showed that variance reduction techniques (SAGA/SVRG) can be used to handle biased gradient estimators in composite stochastic optimization problems, this paper improves the dependence on n from .. to ...'. ---- Clarity & Quality: The paper is written very well and easy to follow. The derived results are clear, and overall presentation excellent (Some concerns remain on the experiments and originality). Originality: It has been shown in previous work [34] that VR methods can be applied to optimization problems of the same structure as studied in this paper. It has been shown that SARAH/SPIDER type VR methods reduce the dependence on n (number components in g) from O(n^(2/3)) to O(n^(1/2)). Thus, the findings of this papers are not surprising and do not contribute novel insights. Further, it appears most of the intro/motivation is taken from [34]. Perhaps a simple reference would be better suited here. Significance: The derived results are certainly worth to be published, though in view of the similarities to [34] I would like to refer to the concerns stated above. The experiments do not reveal much in my opinion, as the results (could) heavily depend on the chosen stepsizes and (sub-)optimal constants in the (parameter-heavy) algorithms. - For instance, the theory seems to indicate that CIVR converges at faster rate than C-SAGA, though this is not the case in the top-right plot. - Batch sizes of the method are set ‘according to theory’. It would be interesting to see if the results change if the batch sizes are tuned as well (i.e. is the suboptimality of VRSC only due to sub-optimal analysis, or inherit to the algorithm)? - The experiments in the Appendix look a bit suspicious as well, as e.g. the batch size has only been tuned for the novel scheme (CIVR-b1), but not for e.g. C-SAGA that often performed better.

Reviewer 3



*** Post rebuttal I have read the rebuttal and the other reviews. The author addressed my concerns, thus I keep my score unchanged. **** The paper presents a generalization of the variance reduction technique for solving the stochastic composite optimization problem of the form f( E[g(x, xi)] ) + r(x). Although not classical in stochastic optimization, this model can be seen for portfolio optimization with risk aversion or reinforcement learning. The core of the algorithm consists of the stochastic approximation of the derivative of the second term. Shortly, the technique maintains two sequences, y_i^t and z_i^t, that estimate g(x_i^t) and it's derivative using a variance reduction technique. The descent direction is thus computed using the formula [z_i^t]^T f(y_i^t), which is a biased estimate of the gradient of the function. The algorithm is then restarted after every epoch (or less, depending on the regularity of the function). The whole algorithm can also be restarted again if the function is strongly convex or gradient-dominated, and thus can potentially have a linear rate of convergence on those problems. It is the first time I see such setting in stochastic optimization, but the motivation paragraph did convince me that studying this case is impactful enough for the machine learning community, even if it looks a bit narrow. Globally, the paper is well written. The authors put some effort to present the results clearly, which is appreciated. I suggest to the author to write somewhere a table showing the different settings and their associated parameters for Algorithm 1. Moreover, the contribution of the paper seems strong. The authors show that in many studied cases, they recover the right rate of convergence, but their analysis is also valid for new settings. Overall, this is a good paper, and I recommend acceptance. I have a few questions/remark for the results presented in the paper. - In section 4.2, restart is required to ensure the fast convergence rate. It took me a bit of time to understand that because the parameters S/eta/T/tau/B seems independent of t. In the end, I finally realized that epsilon was changing over time. I suggest making this point more explicit in this section, for instance, by writing epsilon_t instead. - Again in section 4.2, I feel like the restart on top of a two-loop algorithm seems a bit complicated to implement in practice. Besides, I feel like it may be possible to avoid restart by using time-varying parameters S/eta/T/tau/B. D you this it is possible to prevent the restart strategy in that case? - For all the methods, especially for section 4.2, A sensitivity analysis for the parameter eta can be useful. In practice, the strong convexity constant or the gradient-dominated constant are not well specified. Thus this can slow down the algorithm. - For Katyusha and SVRG, there exists a loopless version that makes the theoretical analysis more straightforward, and this also improves empirical results. (See this paper on arxiv: Don't Jump Through Hoops and Remove Those Loops: SVRG and Katyusha are Better Without the Outer Loop, Dmitry Kovalev, Samuel Horvath, Peter Richtarik). In your case, I think this may help to simplify the algorithm as well as its theoretical analysis. As far as I understand this paper, it consists of changing the deterministic inner-loop into a stochastic one. To you think this approach can be applied to your algorithm?

[Author Response · NeurIPS 2019]

We thank the reviewers for their comments and suggestions. We address their comments separately.

**Reviewer #1:**

*1.1 It is unclear how the bias is handled and the relation to inexact gradient method.*

Reply: First, in this paper we are handling the mean-squared error of estimators, which can be decomposed as $\mathbf{E}[\|\tilde{\nabla}F(x_i^t) - \nabla F(x_i^t)\|^2] = \|\nabla F(x_i^t) - \mathbf{E}[\tilde{\nabla}F(x_i^t)]\|^2 + \mathbf{E}[\|\tilde{\nabla}F(x_i^t) - \mathbf{E}[\tilde{\nabla}F(x_i^t)]\|^2]$. The first term is the squared norm of bias (nonzero in our case) and the second term is the variance. Therefore, our procedure is controlling both of them (see proof of Lemma 1 in Appendix). We shall emphasize this in revision. Second, our method can be considered as an inexact gradient method in the general sense. However, the additive inexactness are carefully controlled by the amount of descent to yield desirable complexity, which is closely related to the property of the estimator. Thus we cannot directly use the standard analysis of inexact gradient method, which will lead to worse complexity.

*1.2 The step size of the algorithm needs to be small in typical non-convex optimization, usually depends on $\epsilon$.*

Reply: Even for non-convex optimization, constant step size can be used if the objective function is smooth. Small or diminishing step sizes are mostly required in stochastic optimization to combat noise in the stochastic gradients, for both convex and non-convex optimization. We are able to use constant step size in the stochastic optimization setting because of much stronger assumption: we assume each realization of $g_\xi(\cdot)$ is a smooth function, not merely its expectation $E_\xi[g_\xi(\cdot)]$ as in classical stochastic optimization. This assumption is key to effective variance reduction in stochastic optimization, and is satisfied in most machine learning problems. We will elaborate on it in the revision.

**Reviewer #3:**

*3.1 Significance of improvement and reference to [34].*

The problems we consider are the same class of problems addressed in [34], thus lead to similar motivation and introduction. We will revise the introduction to be more concise and rely on reference to [34]. While [34] first showed that variance reduction techniques (SAGA/SVRG) can be used to handle biased gradient estimators in composite stochastic optimization problems, this paper improves the dependence on $n$ from $n^{2/3}$ to $n^{1/2}$ using a different estimator SARAH/SPIDER. From theoretical perspective, the improvement is significant for large $n$. More importantly, it reaches the lower bound for the considered problem class in the inner finite-sum case (see next point).

*3.2 Discussion of lower bound in more depth than line 99.*

Reference [8] showed that for nonconvex smooth finite-sum optimization problems (without the outer composition), the lower bound on sample complexity of the component gradients is $O(n^{1/2}\epsilon^{-1})$. That is a special case of the composite problem we consider, where the inner smooth mappings are functions (range dimension is 1) and the outer composition is the identity function. From this perspective, our result also reaches the lower bound for the more general nonlinear composite problems. We will clarify further in the revision, especially with the additional page allowed if accepted.

*3.2 Questions on the experiments.* We agree with the reviewer that more exhaustive search and tuning of parameters may be needed to compare the bests of different algorithms. However, the main purpose of our experiments are to demonstrate some basic behaviors of the algorithms on a few simple examples. Remember that the complexities cited for different algorithms are their theoretical upper bounds. It is not clear if SVRG or SAGA based estimators can achieve the same complexity as CIVR. At least in the convex finite-sum case, their complexities are the same. Nevertheless, we will do some additional experiments to gain more understanding.

**Reviewer #4:**

*4.1 On preventing restart.* First, our restart scheme in Section 4 does not use different $\epsilon$ at different periods (outer loop). We use the desired $\epsilon$ to set all algorithmic parameters once and do not change later. The only reason we use restart is that the output of Algorithm 1 is chosen randomly from all past iterates within one period, which we need to use to start the next period for analysis (see Proof of Theorem 5 in Appendix C.1.) If we can use the last iterate of each period, then there is no need to perform "restart". Nevertheless, it can be avoided by using pre-generated stopping times. Note that we can predetermine period length $T$ and epoch length $\tau$ ($\tau_1 = \cdots = \tau_T$). The output is drawn uniformly from the $T\tau$ iterates. Therefore, we can first uniformly randomly generate the "stop time" $(\bar{i}, \bar{t})$. Then as soon as the algorithm arrives this time, we "restart" seamlessly. We shall add this remark to the revision.

*4.2 The sensitivity on $\eta$.* Our bounds on $\eta$ in the theorems are to guarantee convergence and order of complexity. Smaller $\eta$ can be used, but will encounter a performance penalty, which is explicit in Theorem 1, 2 and 3 on the bounds in (22), (23) and (24) respectively. For theorems in Section 4, small $\eta$ will make $T \propto 1/\eta$ larger (also explicit in the theorems). This will translate into total iteration cost, which we shall clarify in the revision.

*4.3 The "loopless" version (for inner loop, as the outerloop for restart is addressed in 4.1).*

We thank the reviewer for pointing out this interesting literature. According to our analysis, this technique can indeed by applied, and we also get the same complexity. However, different from the convex problem in the "loopless" paper, in the non-convex case the "loopless" proof is a bit more complicated then the current analysis. We will point out this potentially interesting variant of the our method in the revised version.

[Meta-Review · NeurIPS 2019]

There is a clear consensus among the reviewers that the contribution is interesting and that the paper is suitable to publication. The area chair agrees with their assessment and follows their recommendation.